# A selective and augmentable butyrate-FFAR2 signal circuitry programs the cellular identity of enteroendocrine L-cells
Aanya Hirdaramani [1,2], Chia-Wei Cheng [3,4], Aylin C. Hanyaloglu [2] ✉ & Gary Frost [1] ✉

Activation of free fatty acid receptor 2 (FFAR2) on enteroendocrine L-cells mediates secretion of glucagon-like peptide 1 (GLP-1) and peptide YY (PYY), key regulators of central appetite control with therapeutic relevance to obesity. Here, we show that butyrate, a metabolite derived from fermentation of dietary fibre and an FFAR2 agonist, stimulates a PYY-biased profile in a human L-cell model at the transcriptional, morphological and secretory level via an FFAR2-Gαi axis that does not require dynamin-dependent receptor internalisation. We observe that butyrate modulates active Notch cascades within a Hes1-GFP mouse organoid model, which are antagonistic to secretory differentiation, and identify butyrate-dependent regulation of late-stage human enteroendocrine maturation markers, *NeuroD1* and *Pax6*. Butyrate-mediated upregulation of *Pyy* and *Pax6* is enhanced by the FFAR2-selective Gαi biased allosteric agonist AZ-1729. Our study reveals functions of spatiotemporally regulated butyrate-activated FFAR2 signalling mechanisms that could be pharmacologically amplified to fine-tune L-cell populations in the human colon.

Increased dietary fibre intake has been associated with a reduced risk of obesity and metabolic disease such as type 2 diabetes[1]. A key mechanism that has been characterised in driving this protective effect is the enhanced secretion of the satiety-inducing gut hormones, Glucagon-like peptide 1 (GLP-1) and Peptide YY (PYY) following high fibre consumption[2–4]. GLP-1 and PYY are released by enteroendocrine cells predominantly enriched in the distal gastrointestinal tract known as 'L-cells', in response to a range of nutrient and bacterial stimuli, including short chain fatty acids (SCFAs)[5,6]. SCFAs are metabolites derived from the anaerobic bacterial fermentation of dietary fibre by the gut microbiome and are both an energy source and ligands of G-protein coupled receptors (GPCRs) free fatty acid receptor 2 (FFAR2) and 3 (FFAR3)[7,8]. Acetate, propionate and butyrate are the predominant SCFAs in the human gut and activate human FFAR2 and FFAR3 with distinct potencies (human FFAR2; Ace = Pro > But, FFAR3; Pro = But > Ace)[7,8]. FFAR2 exhibits dual coupling capabilities to Gαi and Gαq heterotrimeric G protein pathways, whereas FFAR3 predominantly activates Gαi signalling[7,8]. While targeted colonic delivery of propionate-inulin ester has known beneficial actions in regulating appetite and weight

gain in humans[9], FFAR2 has been only demonstrated to be the primary receptor in driving GLP-1 and PYY release in response to SCFA activation in rodent models[10–12].

The mechanisms driving acute anorectic gut hormone release by propionate-activated FFAR2 have implicated both calcium and Gαi/p38 signalling in driving acute GLP-1 release from mouse L-cells[10,12]. However, SCFAs mediate multiple functions, beyond acute anorectic gut hormone release, not only across cell types but within L-cells[13–15]. How each SCFA elicits distinct physiological functions from FFAR2/3 that share common upstream G protein signal machinery is poorly understood[11]. One mechanism GPCRs employ is spatial pleiotropy in signalling, which for numerous GPCRs has been shown to be integral in achieving diversity and precision in their ligand-induced physiological effects[16]. We have previously demonstrated a key role for FFAR2 internalisation and endosomal localisation for propionate-mediated Gαi/p38 signalling in mouse L-cell models and colonic crypts[12].

In addition to hormone secretion, the response of L-cells to intestinal metabolites also influences their differentiation trajectories within the

[1]Section of Nutrition, Department of Metabolism, Digestion and Reproduction, Imperial College London, London, UK. [2]Institute of Reproductive and Developmental Biology, Department of Metabolism, Digestion and Reproduction, Imperial College London, London, UK. [3]Columbia Stem Cell Initiative, Columbia University Irving Medical Center, New York, NY, USA. [4]Department of Genetics and Development, Columbia University Irving Medical Center, New York, NY, USA. ✉e-mail: a.hanyaloglu@imperial.ac.uk; g.frost@imperial.ac.uk

intestinal stem niche[17]. For example, L-cells derived from mice fed a high-fat diet have been shown to exhibit downregulated expression of nutrient-sensing machinery and enteroendocrine-specific transcription factors[18]. In contrast, supplementation with SCFAs has upregulated transcription factor cascades involved in L-cell development, and increased L-cell number in mouse and human organoids[19]. In human studies, this inherent 'plasticity' in L-cell differentiation has been demonstrated in obese individuals who exhibit a depleted density of GLP-1 and PYY-positive EECs that is restored upon laparoscopic sleeve gastrectomy[20,21]. EECs, like all intestinal cell types, are derived from the differentiation of multipotent stem cells. The primary checkpoint for a stem cell progenitor destined for an EEC fate is antagonised by the Notch effector protein Hes1[22]. Hes1 has been found to be upregulated in the obese intestine and similarly reversed upon weight loss surgery[20]. Alongside local signalling gradients such as Wnt, Notch and BMP that co-ordinate the ISC niche, several GPCRs have been shown to regulate intestinal cell proliferation and differentiation[23]. SCFA-mediated activation of FFAR2 has been linked to increased numbers of PYY-positive L-cells by employing knockout mouse models[24], with indication that the SCFA butyrate may drive PYY transcription through poorly understood receptor and non-receptor mechanisms[25]. The signalling cascades and transcriptional function(s) of butyrate-mediated FFAR2 signalling in the human L-cell, remains unclear.

Here, we demonstrate that in a human L-cell model, SCFAs activate FFAR2 signalling pathways that vary highly with chain length of the fatty acid and exhibit differential location bias. This location bias in G protein signalling in turn mediates the ligand-specific actions of each SCFA on downstream L-cell functions, specifically at a transcriptional, morphological and endocrine level. Furthermore, we demonstrate that the SCFA butyrate promotes Notch-sensitive enteroendocrine differentiation to a 'PYY-bias' phenotype that can be augmented via a selective FFAR2 allosteric biased ligand, highlighting a critical role for butyrate in driving specific steps of L-cell differentiation via a spatially regulated FFAR2-Gαi axis.

## Results

### The SCFA butyrate induces distinct endocrine and morphological changes in the human colonic enteroendocrine NCI-H716 cell line

The NCI-H716 cell line is a widely used model of human L-cells, particularly in studying nutrient-stimulated GLP-1 and PYY secretion[26,27]. We confirmed gene expression of both SCFA GPCRs *Ffar2* and *Ffar3* in this cell line, alongside GLP-1 precursor proglucagon (*Gcg*) and *Pyy* (Fig. S1a). In addition, previously reported[25] SCFA-mediated effects on *Gcg* and *Pyy* gene expression were confirmed (Fig. 1a). NCI-H716 cells were treated with 2 mM of acetate, propionate or butyrate for 24 h. Propionate and butyrate, but not acetate, significantly decreased *Gcg* expression. In contrast, all three SCFAs significantly increased *Pyy* transcript abundance, and this effect positively correlated with chain length (acetate (C2) < propionate (C3) < butyrate (C4)) with butyrate stimulating the largest increase in expression. This dramatic increase induced by butyrate on *Pyy* transcript was also reflected at the secretory level, however acetate and propionate did not increase PYY secretion in NCI-H716 cells (Fig. 1b). Using immuno-fluorescent staining and confocal imaging paired with adaptive deconvolution, achieving a resolution limit of 120 nm, GLP-1 and PYY granules were resolved to distinct populations in NCI-H716 cells (Fig. S1b–d). In untreated cells, PYY granules were less abundant than GLP-1 granules, on average comprising $19.01 \pm 2.138\%$ of the total granule (GLP-1 and PYY) pool per cell (Fig. 1c, d). Following treatment with butyrate but not acetate or propionate, average PYY granule numbers per cell were significantly increased; PYY granules in butyrate-treated cells also comprised a significantly higher proportion of the total granule pool ($34.26 \pm 4.456\%$) compared to untreated cells (Fig. 1e).

In addition, NCI-H716 appeared to undergo a prominent morphological shift upon treatment with butyrate, exhibiting elongated morphologies characteristic of specific L-cell populations enriched in the human distal intestine[28,29] (Fig. 1f). PYY-immunoreactive cells in primary human colonic tissue have been shown to often adopt a tall anatomy with extended cytoplasmic processes, weaving between cells from the lamina to the gut lumen[28,29]. To quantify the potential effects of SCFA treatment on inducing an L-cell like morphology, we measured cellular area, feret diameter (defined as the longest distance across the cell in any direction), and circularity (Perimeter$^2$/$4\pi \times$ Area) of cells using a CellMask stain (Fig. 1g). We measured a significant increase in feret diameter amongst propionate- and even more so amongst butyrate-treated cells. Cellular area was also significantly increased by butyrate treatment, further confirming the appearance elongated shapes we had observed in these populations. Acetate had no prominent effect on NCI-H716 morphology. Together these findings suggest that butyrate not only functionally drives an L-cell line towards production of PYY at a transcriptional, translational and secretory level, but also induces a morphology akin to PYY-immunoreactive L-cells in the colon[28,29].

### Butyrate activates distinct G-protein signalling profiles compared to acetate and propionate via FFAR2 in human L-cells

To uncover the mechanistic pathways underlying the reprogramming actions of butyrate in NCI-H716 cells, we characterised and compared the FFAR2/3G protein signalling profiles following treatment with each SCFA. FFAR2 can signal via Gαi and Gαq pathways. Activation of Gαq stimulates the activity of phospholipase C-β, which cleaves PIP2 to produce DAG and IP3 leading to release of calcium from intracellular stores. We investigated SCFA-mediated activation of Gαq in the human NCI-H716 cells by measurement of IP1 levels, a downstream metabolite of IP3 (Fig. 2a). We observed concentration-dependent activation of Gαq signalling following stimulation with either acetate (pEC50 = $3.256 \pm 0.970$) or propionate (pEC50 = $3.392 \pm 0.962$), which was inhibited by the FFAR2-selective orthosteric antagonist GLPG0974[30] (Fig. 2b). In comparison, butyrate stimulated a weaker increase in intracellular IP1 levels (pEC50 = $2.796 \pm 0.766$). Gαq activation profiles of SCFAs in NCI-H716 cells were also assessed by measurement of intracellular calcium mobilisation (Fig. 2c, d). Acetate and propionate triggered a rapid, monophasic increase in intracellular calcium. Butyrate failed to induce significant calcium mobilisation, supporting minimal/weak activation of Gαq signalling in comparison to acetate and propionate.

Activation of Gαi signalling by GPCRs inhibits adenylyl cyclase activity, thereby downregulating the conversion of ATP to cAMP and activity of cAMP-dependent protein kinase (PKA). NCI-H716 cells were treated with the adenylyl cyclase activator forskolin, and the effect of SCFAs on forskolin-mediated cAMP accumulation was measured. All three SCFAs exhibited a significant inhibition of forskolin-mediated cAMP accumulation in a concentration-dependent manner, with butyrate (pIC50 = $-3.348 \pm 0.654$) exhibiting significantly lower potency than propionate (pIC50 = $-4.157 \pm 0.745$) and acetate (pIC50 = $-4.191 \pm 0.674$) (Fig. 2e). Given that both FFAR2 and FFAR3 are Gαi/o-coupled and both receptors are detected at mRNA level, it was important to resolve the mechanistic involvement of each receptor in driving Gαi signalling. The selective FFAR2 antagonist GLPG0974 inhibited the Gαi signalling across all doses of all three SCFAs employed, although propionate-induced an unexpected increase in cAMP signals following treatment with GLPG0974. Overall, these data suggest that FFAR2 rather than FFAR3 mediates SCFA-driven Gαi responses in NCI-H716 cells (Fig. 2f).

### Butyrate and a selective FFAR2 allosteric agonist increase *Pyy* secretion in an internalisation-independent manner

We have previously demonstrated that in mouse enteroendocrine L-cells and colonic crypts, propionate-stimulated GLP-1 release is dependent on the internalisation of FFAR2 for Gαi signalling[12]. An understanding of SCFA-encoded trafficking profiles of human FFAR2 is limited and could be crucial to unravelling functions specific to FFAR2 activation by butyrate in human L-cells. NCI-H716 cells were transfected with a FLAG-tagged human FFAR2 (hFFAR2), and receptor internalisation was visualised by 'feeding' live cells with anti-FLAG antibody to specifically label cell surface receptor to track its endocytosis, followed by confocal

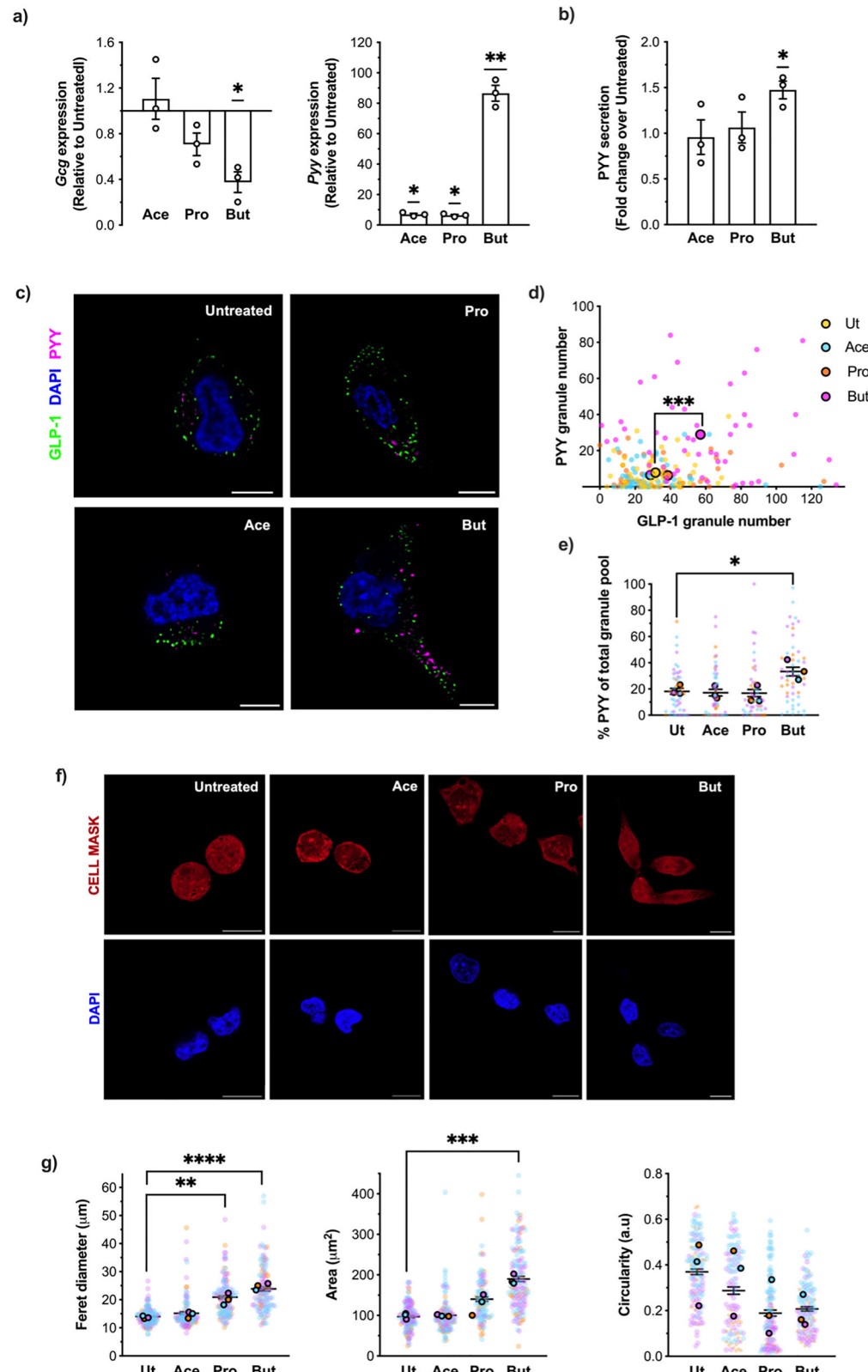

microscopy imaging. Under basal conditions, receptor was primarily at the plasma membrane with some endosomes apparent indicating constitutive internalisation, as observed for mouse FFAR2[12]. However, treatment with acetate, propionate or butyrate resulted in an increase in the abundance of FFAR2-positive endosomes (Fig. 3a). Both constitutive and propionate-induced internalisation of mouse FFAR2 proceeds via a

dynamin-dependent mechanism[12], in which the GTPase dynamin facilitates the intracellular scission of caveolae and clathrin-coated pits at the plasma membrane. We confirmed both constitutive and SCFA-induced trafficking of hFFAR2 in NCI-H716 cells is also dynamin-dependent as pre-treatment with dynamin inhibitor Dyngo-4a inhibited receptor internalisation (Fig. 3a).

**Fig. 1 | Butyrate enhances *Pyy* expression, PYY secretion and granule number and induces morphological changes in NCI-H716 cells.** NCI-H716 cell were treated with either 2 mM of sodium acetate (Ace), sodium propionate (Pro) or sodium butyrate (But) for 24 h. **a** *Gcg* and *Pyy* transcript expression in NCI-H716 cells were detected by RT-qPCR. Gene expression relative to untreated was determined using the $2^{-\Delta\Delta Ct}$ method with ribosomal protein L12 (*Rpl12*) as a housekeeping gene control. Data are plotted as the mean ± SEM of fold-change ($2^{-\Delta\Delta Ct}$) values. Symbols depict the mean ± SEM from $n = 3$ independent repeats. Statistical significance was assessed on ΔΔCt values. (**$p < 0.01$; ****$p < 0.0001$, One-way ANOVA with Dunnett's post-hoc test vs Untreated). **b** Secretion of PYY from NCI-H716 cells was assayed by ELISA. Following SCFA treatment as in (**a**), cells were incubated with ligand-free secretion buffer for 2 h and supernatants were subsequently assayed for PYY concentration by ELISA. Data was normalised as fold change over untreated cells. Symbols depict the mean ± SEM from $n = 3$ independent repeats (*$p < 0.05$, One-way ANOVA with Dunnett's post-hoc test vs Untreated). Mean PYY secretion in untreated supernatants across repeats was 106.6 ± 8.196 pM. **c** Representative images of fixed NCI-H716 stained with anti-GLP-1 and anti-PYY antibodies captured by confocal microscopy in super-resolution by an adaptive deconvolution module (LIGHTNING, Leica). Scale bar = 10 μm. Conditions were imaged across $n = 3$ independent experiments. (**d**) PYY and GLP-1 granule numbers in Untreated (Ut) or SCFA-treated NCI-H716 cells in super-resolved confocal images. Larger data points represent the mean ±

SEM from $n = 3$ independent repeats, coloured by treatment and values from individual cells are shown as smaller data points with corresponding colours to treatment. $n = 50$ cells were collectively analysed across repeats. (***$p < 0.001$, One-way ANOVA with Dunnett's post-hoc test vs Untreated for PYY). **e** Data from (**d**) presented as % PYY granules of total (GLP-1 + PYY) in Untreated (Ut) or SCFA-treated NCI-H716 cells. Larger data points represent the mean ± SEM of three independent experiments, coloured by individual biological repeats. Values from individual cells are shown as smaller data points with colours corresponding to the respective experimental run. (*$p < 0.05$; One-way ANOVA with Dunnett's multiple comparisons; Ut vs Ace/Pro/But). **f** Representative images of fixed NCI-H716 cells captured via confocal microscopy incubated with HCS CellMask Deep Red Stain (stains across the cytoplasm and nuclei) and DAPI nuclei stain. Scale bar = 10 μm. Conditions were imaged in at least across $n = 3$ independent experiments. **g** Measurements of area, circularity and feret diameter obtained by automatic particle analysis on thresholded images of Untreated (Ut) and SCFA-treated NCI-H716 cells by ImageJ. Larger data points represent the mean ± SEM from $n = 3$ independent experiments, coloured by individual repeats. $n = 120$ cells were collectively analysed, and values from individual cells are shown as smaller data points with colours corresponding to the respective experimental run (****$p < 0.0001$; ***$p < 0.001$; *$p < 0.05$, One-way ANOVA with Dunnett's multiple comparisons; Ut vs Ace/Pro/But). See also Fig. S1.

To uncover the role of SCFA-induced hFFAR2 internalisation on receptor signalling profiles in NCI-H716 cells, Gαq and Gαi signalling was measured following treatment with Dyngo-4a. A strong inhibition of both acetate- and propionate-stimulated increases in IP1 accumulation by Dyngo-4a was observed, indicating that internalisation of hFFAR2 was required for full activation of Gαq (Fig. 3b). Although butyrate's ability to activate Gαq is limited, the small increase in IP1 levels was also inhibited by Dyngo-4a. SCFA-mediated activation of Gαi signalling was only partially inhibited by Dyngo-4a (Fig. 3c). These findings demonstrate that hFFAR2 exhibits a differential requirement for internalisation for signalling via each G protein pathway by SCFAs.

As butyrate activates primarily Gαi signalling via an FFAR2-dependent mechanism in NCI-H716 cells, potentially from both the plasma membrane and following receptor internalisation, we next assessed the mechanistic involvement of this pathway in driving expression of *Pyy*. Following pre-treatment with the FFAR2 antagonist GLPG0974, butyrate-mediated *Pyy* upregulation was partially but significantly inhibited (Fig. 3c). Likewise, pre-treatment with the Gαi inhibitor pertussis toxin (PTX), at a concentration which we had observed to significantly diminish Gαi responses activated by up to 10 mM of butyrate (Fig. S2a), partly but significantly inhibited butyrate-mediated Pyy gene upregulation. In contrast, treatment with an inhibitor of Gαq, YM-275890, did not affect butyrate-mediated increases in *Pyy* transcript, supporting the involvement of an FFAR2-Gαi axis in this downstream response. Although FFAR2-Gαi signalling in NCI-H716 cells was partially dependent on receptor internalisation, Dyngo-4a treatment did not inhibit butyrate-mediated *Pyy* upregulation, indicating a plasma membrane hFFAR2-Gαi signal in driving this response. In contrast to our findings on *Pyy* expression, butyrate-stimulated PYY secretion was not inhibited by GLPG0974 and signalling inhibitors, indicating a potential FFAR2-independent mechanism. (Fig. S2b).

To further investigate a role for an FFAR2-Gαi pathway in driving *Pyy* expression, we employed a selective hFFAR2 allosteric biased agonist of Gαi activation, AZ-1729[31]. Stimulation of NCI-H716 cells with AZ-1729, with or without butyrate, resulted in activation of Gαi signalling and potentiation of butyrate responses, confirming its ability as an agonist and positive allosteric modulator (Fig. S3). In NCI-H716 cells expressing FLAG-hFFAR2, we observed dynamin-dependent receptor internalisation following treatment with AZ-1729, which was blocked by Dyngo-4a pre-treatment (Fig. 4a). Dyngo-4a did not, however, significantly impact AZ-1729-mediated Gαi signalling in NCI-H716 cells, suggesting that the FFAR2-Gαi signal pathway activated by AZ-1729 is internalisation-independent and occurs at the plasma membrane (Fig. 4b). AZ-1729 stimulated a ~2-fold increase in *Pyy* expression relative to baseline, however when co-treated with butyrate

significant increased *Pyy* mRNA levels a further ~2 fold over the ~16 fold response driven by butyrate, suggesting a synergistic action (Fig. 4c, d). Together, these findings not only reinforce the mechanistic involvement of a plasma membrane-localised hFFAR2-Gαi signalling pathway in *Pyy* expression but that an allosteric modulator can amplify the actions of butyrate in L-cells.

## Butyrate modulates NOTCH activity to direct L-cell maturation

In addition to modulating the endocrine functionalities of L-cells in the intestinal milieu, SCFAs have been shown to influence EEC differentiation[19,24]. We next determined if the transcriptional, endocrine and morphological actions of butyrate reflect activation of key differentiation pathways driving L-cell development. In the intestinal stem cell niche (ISC), secretory lineage differentiation is antagonised by Notch. The Notch effector protein Hes1 is expressed in different cellular populations in the ISC niche where it serves distinct functions; in ISCs its promotes their continuous proliferation, whereas in progenitors cells it encourages differentiation to an absorptive (i.e. enterocyte) rather than a secretory fate (i.e. goblet, Paneth, EEC) via repression of the master secretory lineage development transcription factor, Atoh1 (Fig. 5a)[32–34]. To investigate the role of butyrate on secretory lineage differentiation, we established colonic organoids from a previously described Notch reporter Hes1-GFP mouse[33].

Organoid cultures were established from isolated Hes1-GFP colonic crypts and comprised of a mixture of two canonical organoid populations with heterogeneous morphologies and cellular makeup; a 'budding' population and a 'cystic' population (Fig. 5b). Mature 'budding' organoids consist of projections of crypt-base-like domains, enriched in proliferative stem cells, and upper-crypt-like domains in which late-stage progenitors and mature differentiated cells of both absorptive and secretory lineages reside. As such, budding organoids recapitulate the in vivo architecture and functional compartmentalisation of stem cell and differentiated zones in the intestinal colonic crypt[35–37]. In contrast, 'cystic' organoids consist of a lumen surrounded by a thin epithelial monolayer of stem cells akin to populations of continuously dividing ISCs that occupy the base of crypts and support homoeostatic renewal in the ISC niche[38]. To understand whether differences in Notch activity (i.e. Hes1-GFP fluorescence) would be representative of heterogeneous cell ontologies in this reporter model, we analysed *Hes1* expression at single cell resolution in the mouse colon using the publicly available database Tabula Muris[39] (Fig. S4a). We confirmed *Hes1* expression was high across epithelial and enterocyte populations, and low in EEC and brush cells suggesting that differences in absorptive and secretory lineages could be delineated by reporter fluorescence.

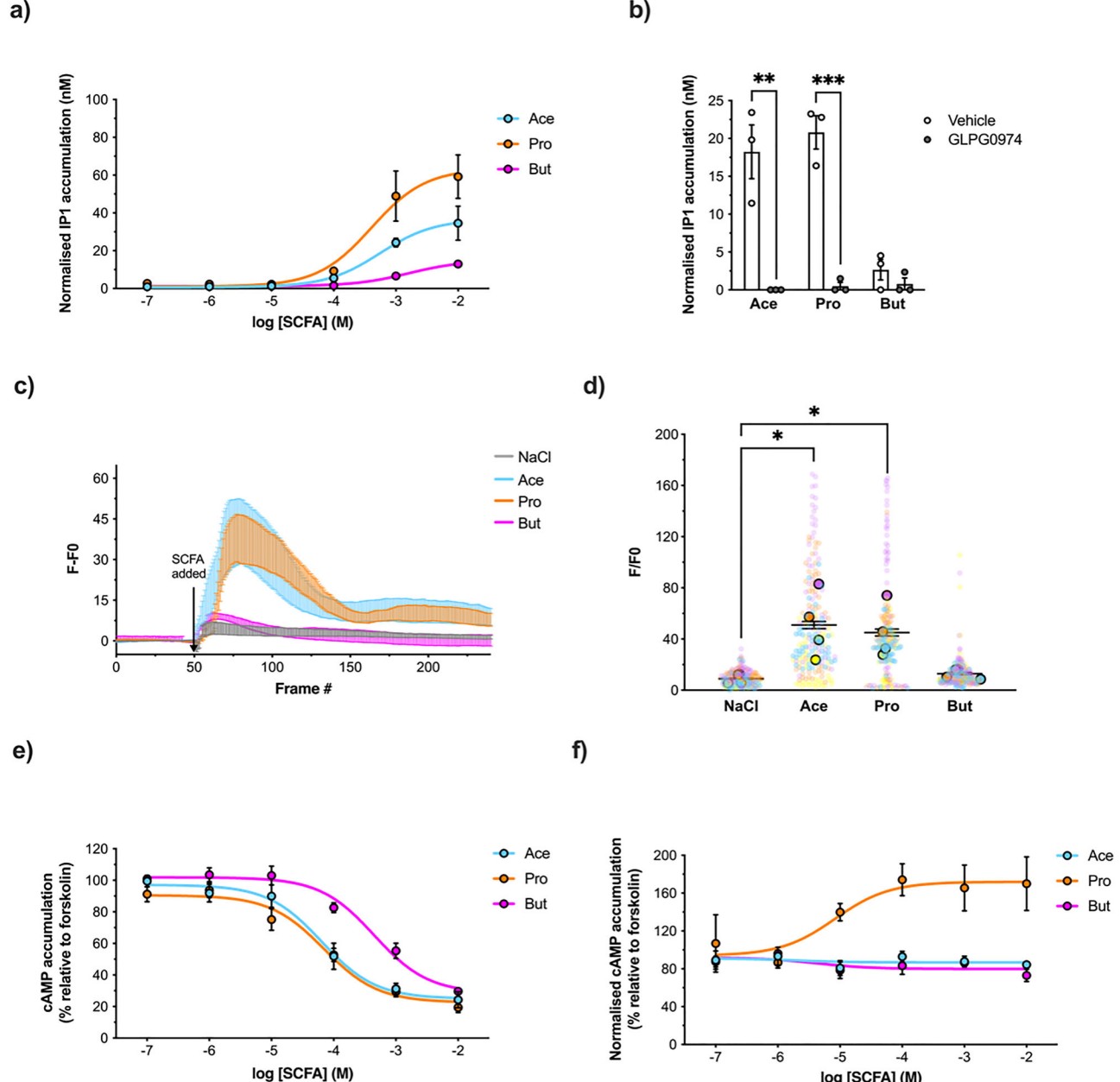

**Fig. 2 | SCFAs stimulate distinct Gαi/o and Gαq/11 signalling profiles in NCI-H716 cells via activation of FFAR2. a** Intracellular IP1 accumulation measured in NCI-H716 cells following 2 h incubation with increasing concentrations of either NaCl, sodium acetate (Ace), sodium propionate (Pro) or sodium butyrate (But). Data were normalised by subtraction of NaCl response at each concentration. Data represent the mean ± SEM from $n = 3$ independent experiments. **b** IP1 accumulation in NCI-H716 cells pre-treated with DMSO (Vehicle) or FFAR2 antagonist GLPG0974 (1 μM, 15 min) followed by 2 h incubation with 1 mM of NaCl, Ace, Pro or But. Data were normalised by an identical method to (**a**). Symbols depict the mean ± SEM from $n = 3$ independent experiments (***$q < 0.001$;**$p < 0.01$, $t$-test Vehicle vs GLPG0974 response with post-hoc Benjamin-Hochberg procedure). **c, d** Intracellular Ca$^{2+}$ increases in NCI-H716 cells measured following 1 h incubation with Fluo-4AM fluorescent dye, Cells were imaged live by confocal microscopy at 1.2 frames/second, at basal for 1 min, and then 10 mM of NaCl, Ace, Pro or But was added. Data are presented as fluorescent intensity normalised to untreated baseline (F-F0). **c** Mean F-F0 traces from the beginning to end of experimental capture. Data are shown as SEM of experimental means. **d** Mean Maximal F-F0 following ligand addition. Large symbols depict the mean ± SEM from $n = 4$ independent experiments, coloured by individual repeats and smaller symbols represent values of individual cells with colours corresponding to their experimental run. $n = 40$ cells were collectively analysed. (*$p < 0.05$, One-way ANOVA with Dunnett's multiple comparisons: NaCl vs Ace/Pro/But). **e** Inhibition of intracellular cAMP accumulation in NCI-H716 cells treated with 500 nM 3-isobutyl-1- methylxanthine (IBMX) for 5 min and then stimulated with 3 μM of forskolin (FSK) in the presence of increasing doses of Ace, Pro or But for 5 min. Data are expressed as % response of cAMP accumulation in FSK-treated cells and represent the mean ± SEM from $n = 5$ independent experiments. **f** Inhibition of cAMP accumulation in NCI-H716 cells following pre-treatment with DMSO (Vehicle) or GLPG0974 antagonist (1 μM, 15 min) and subsequent ligand stimulation as in (**e**). Data represent the mean ± SEM from $n = 3$ independent experiments.

We monitored the effect of butyrate on Hes1-GFP fluorescence in budding (Fig. 5c, d) and cystic (Fig. 5e, f) organoids daily across a 72 h period. In budding populations, butyrate stimulated a significant decrease in mean normalised fluorescence at 48 h and 72 h, which was not observed across untreated budding organoids (Fig. 5c). We did not detect time-dependent differences in mean total area of budding organoids in either butyrate or untreated populations across timepoints (Fig. S4b), however, mean areas devoid of GFP signal were significantly increased in butyrate-treated budding organoids across timepoints (Fig. 5d). In cystic organoids, mean GFP fluorescence significantly

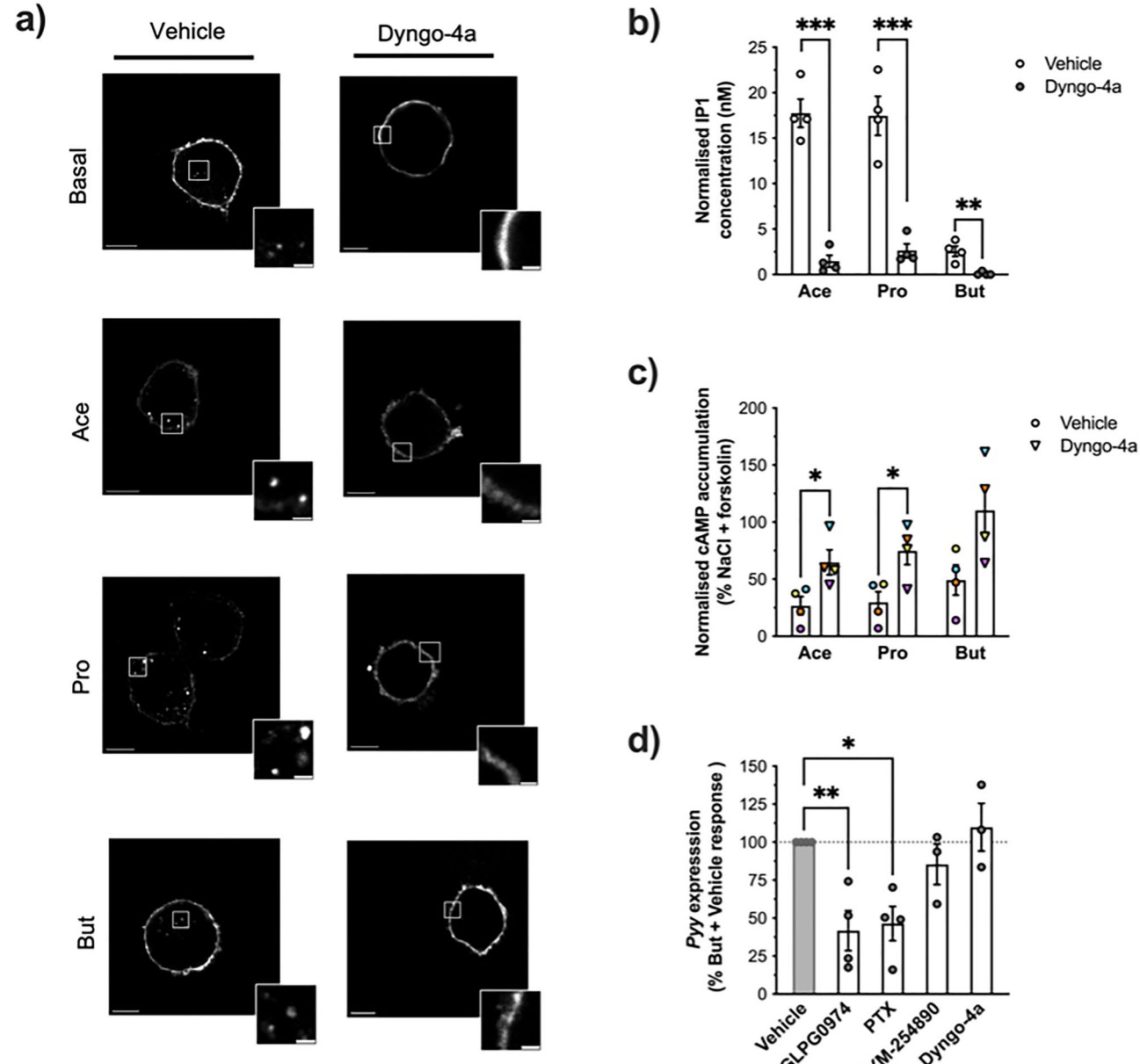

**Fig. 3 | SCFAs stimulate dynamin-dependent internalisation of FFAR2 from the plasma membrane, which modulate receptor signalling profiles. a** Representative confocal images of NCI-H716 cells transiently transfected with FLAG-tagged human FFAR2. Transfected cells were treated for 45 min with DMSO (Vehicle) or 50 µm of dynamin-dependent endocytosis inhibitor Dyngo-4a, followed by staining with anti-FLAG M1 antibody for 30 min. 1 mM of sodium acetate (Ace), sodium propionate (Pro) or sodium butyrate was added to cells in the latter 20 min of staining. $n = 10$ cells were captured per treatment condition across $n = 3$ independent experiments. Scale bar = 10 µm. Inset scale bar = 2 µm. **b** Intracellular IP1 accumulation in NCI-H716 cells measured following 45 min pre-treatment with DMSO (Vehicle) or 50 µM of Dyngo-4a and then a 2 h stimulation with 1 mM of NaCl, Ace, Pro or But. Data were normalised by subtraction of NaCl response at each concentration. Symbols depict the mean ± SEM from $n = 4$ independent experiments. (***$q < 0.001$, **$q < 0.01$, $t$-test Vehicle vs Dyngo-4a response with post-hoc Benjamin-Hochberg procedure). **c** Intracellular cAMP accumulation measured in NCI-H716 cells pre-treated for 45 min with DMSO (Vehicle) or 50 µM Dyngo-4a and then stimulated for 5 min with 500 nM 3-isobutyl-1- methylxanthine (IBMX)

followed by 3 µM of forskolin (FSK) in the presence of 1 mM of NaCl, Ace, Pro or But for 5 min. Data are expressed as % response of cAMP accumulation in NaCl + FSK-treated cells. Symbols depict the mean ± SEM from $n = 4$ independent experiments, with different colours marking individual experimental runs. (*$q < 0.05$, $t$-test Vehicle vs Dyngo-4a response with post-hoc Benjamin-Hochberg procedure). **d** Butyrate-mediated upregulation of *Pyy* transcript expression measured in NCI-H716 cells pre-treated with DMSO (Vehicle), GLPG0974 (1 µM, 15 min), PTX (500 ng/ml, 20 h), Gαq inhibitor YM-254890 (10 nM, 15 min) or dynamin-dependent endocytosis inhibitor Dyngo-4a (50 µM, 45 min) followed by 24 h incubation with 2 mM of But. Transcript levels were detected by RT-qPCR and gene expression relative to untreated was determined using the $2^{-\Delta\Delta Ct}$ method with ribosomal protein L12 (*Rpl12*) as a housekeeping gene control. But + Inhibitor response is shown as % percentage of But + Vehicle response (dotted line at 100%). Symbols depict the mean ± SEM from $n = 3$ independent experiments (**$p < 0.01$, *$p < 0.05$; One-way ANOVA with Dunnett's post-hoc test, But + Vehicle (control) vs But + inhibitor). See also Fig. S2.

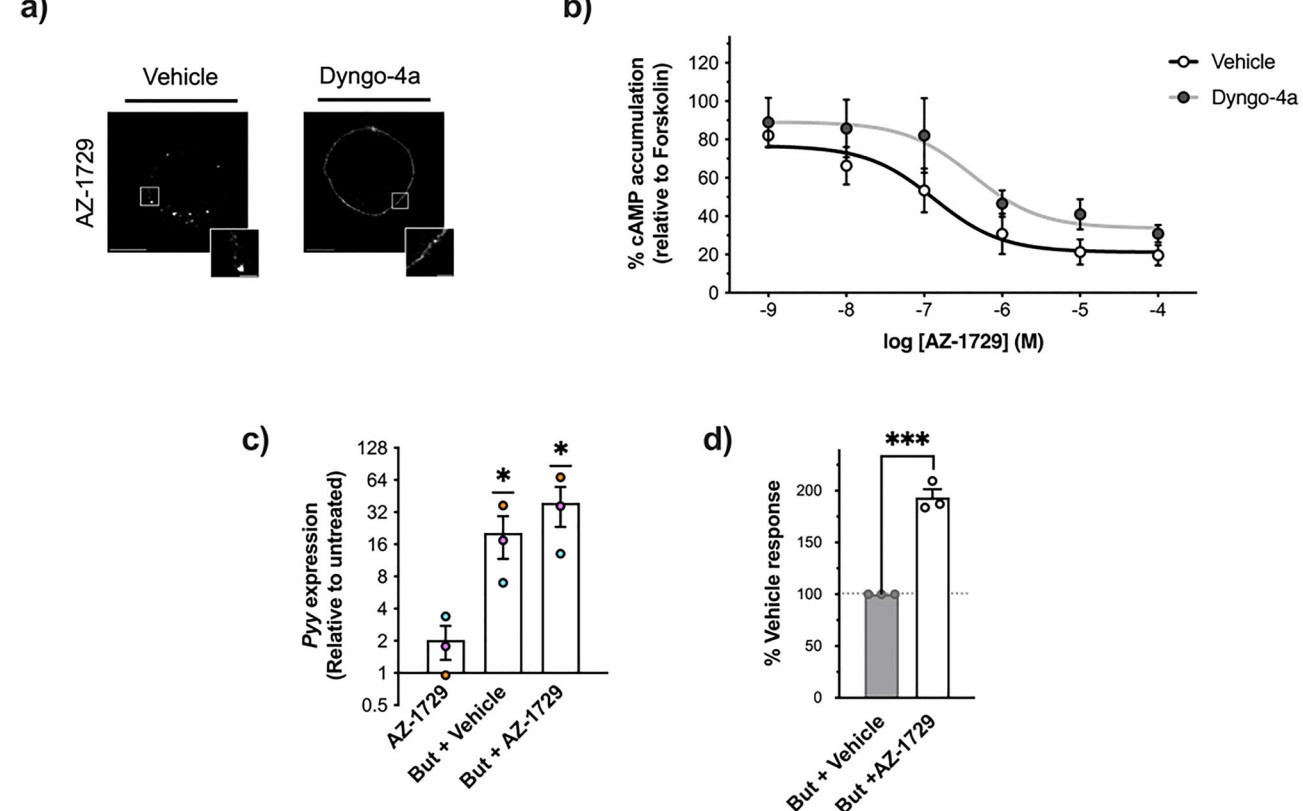

**Fig. 4 | AZ-1729 activates FFAR2-Gαi signalling independent of receptor internalisation to upregulate *Pyy* expression. a** Representative confocal images of NCI-H716 cells transiently transfected with FLAG-tagged human FFAR2. Transfected cells were treated for 45 min with DMSO (Vehicle) or 50 μM of dynamin-dependent endocytosis inhibitor Dyngo-4a, followed by staining with anti-FLAG M1 antibody for 30 min. FFAR2-Gαi biased allosteric agonist AZ-1729 (1 μM, 20 min) was added to cells. *n* = 15 cells were captured per condition across *n* = 3 experiments. Scale bar = 10 μm. Inset scale bar = 2 μm. **b** Intracellular cAMP accumulation measured in NCI-H716 cells pre-treated for 45 min with DMSO (Vehicle) or 50 μM Dyngo-4a and then stimulated for 5 min with 500 nM 3-isobutyl-1-methylxanthine (IBMX) followed by 3 μM of forskolin (FSK) in the presence of 1 μM of AZ-1729 for 5 min. Data are expressed as % response of cAMP accumulation in FSK-treated cells and represent the mean ± SEM from *n* = 3 independent experiments. **c, d** *Pyy* transcript expression in NCI-H716 cells treated for 24 h with 1 μM AZ-1729, 2 mM sodium butyrate (But) + DMSO (Vehicle) or a combination of 2 mM But and 1 μM AZ-1729. Transcript levels were detected by RT-qPCR and gene expression relative to untreated was determined using the $2^{-\Delta\Delta Ct}$ method with ribosomal protein L12 (*Rpl12*) as a housekeeping gene control. Data are plotted as the mean ± SEM of fold-change ($2^{-\Delta\Delta Ct}$) values data. Symbols depict the mean ± SEM from *n* = 3 independent experiments, coloured by individual experimental runs. Statistical significance was assessed on ΔΔCt values. (\*\**p* < 0.01, \*\*\**p* < 0.001, One-way ANOVA with Dunnett's post-hoc test, Untreated vs Ligand(s)). In (**d**), But + AZ-1729 response is shown as % of But + Vehicle response (dotted line at 100%). (\*\*\**p* < 0.01, *t*-test But + Vehicle vs But + AZ-1729). See also Fig. S3.

---

increased in a time-dependent manner in untreated conditions, which was suppressed in butyrate-treated equivalents (Fig. 5e, f). In addition, a significant decrease in mean % GFP-negative area was apparent in untreated cystic organoids at 48 h and 72 h but not detected in butyrate-treated populations (Fig. 5f). Total area of cystic organoids was unchanged across time points in untreated conditions yet was significantly increased following 72 h of butyrate treatment (Fig. S4c). Together, these observations in cystic and budding organoids illustrate an inhibitory effect of butyrate on Notch activity across ISCs and maturing cell types in the mouse colonic epithelium respectively.

Notch pathway activity in mouse colonoid cultures was also investigated by bulk RNA-seq analysis. (Fig. 5g). Pathway enrichment analysis revealed that Notch signalling was downregulated by butyrate; HALLMARK_NOTCH_SIGNALING singscores[40] were lower in butyrate-treated cultures at both 48 h (z-scores; −0.55 butyrate vs 0.66 untreated) and 72 h (−1.11 vs 1.01 untreated, respectively), and were significant across conditions (*q* < 0.0001). These transcriptomic data are consistent with our imaging observations in reporting a sustained inhibition of Notch activity by butyrate in the mouse colon. Next, we performed immunofluorescent staining to explore whether butyrate-mediated Notch inhibition was indicative of maturation towards a Pyy-biased phenotype. We observed minimal numbers of PYY positive cells in organoids, consistent with the scarce abundance of EECs in the ISC niche (<1%), but an upregulation of this population in butyrate-treated samples (Fig. S4d). Overall, our findings in mouse colonoids uncover an inhibitory effect of butyrate on Notch in directing differentiation towards secretory fates including PYY positive populations.

## Butyrate-activated FFAR2 selectively modulates late stage maturation of EECs by increasing Pax6, a pathway amplified by AZ-1729

Next, we explored whether butyrate-mediated inhibition of Notch was indicative of enhanced cellular differentiation to EECs in the human colon, and if this effect was restricted to specific stages of differentiation. *Atoh1* expression marks the initiation of a conserved cascade of transcription factors which are transiently and sequentially expressed, consisting of *Atoh1*, *Neurog3* and then *NeuroD1*[41–44] (Fig. 6a). In NCI-H716 cells, incubation with butyrate significantly downregulated *Atoh1*, whereas limited effect on *Neurog3* transcript levels were detected (Fig. 6b). Notably, butyrate also significantly increased *NeuroD1* expression by ~4 fold. When we compared the effects of butyrate on modulating expression levels of *Atoh1, Neurog3* and *NeuroD1* with acetate and propionate, we only detected a significant decrease in *Atoh1* expression by propionate (Fig. S5).

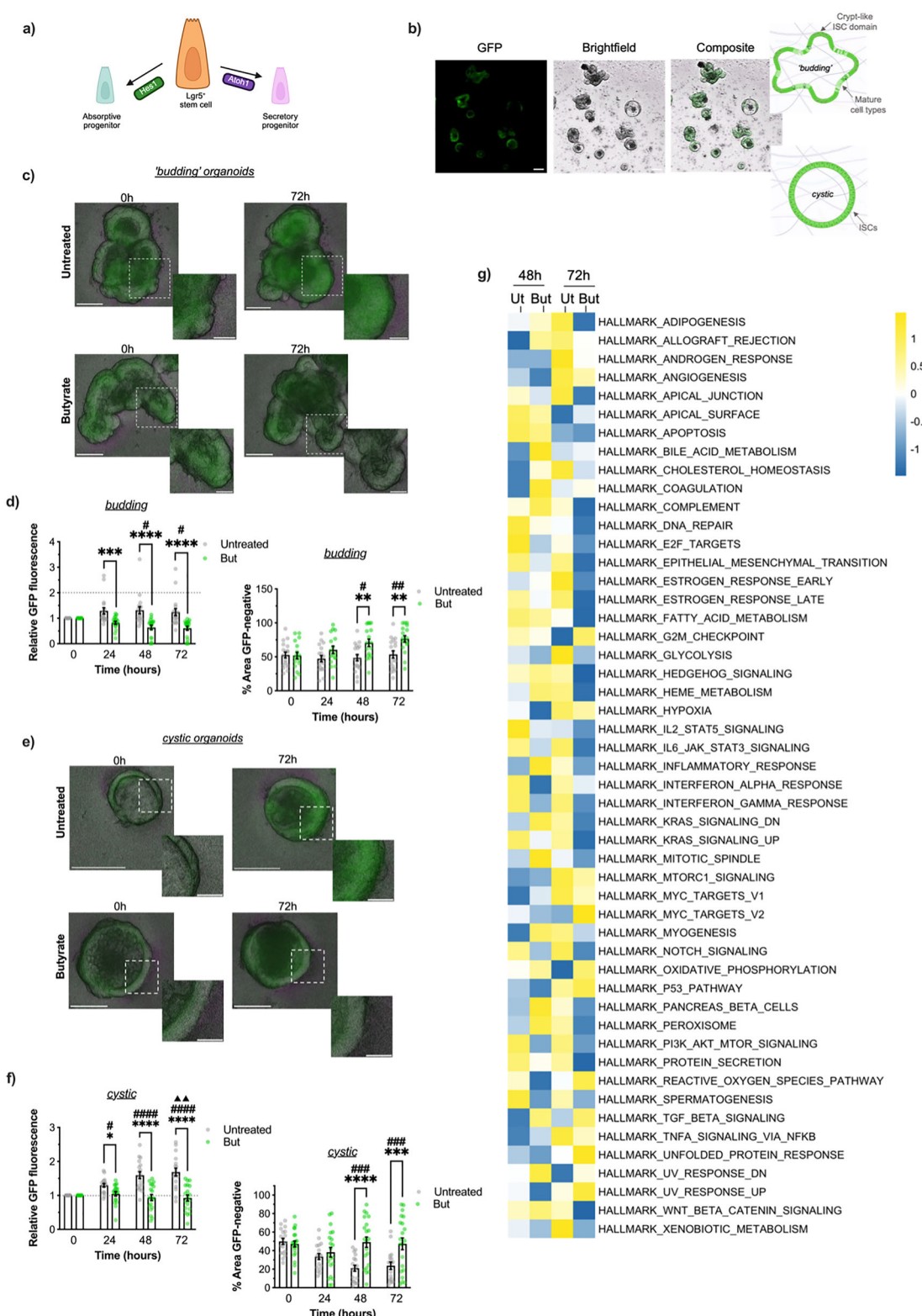

Downstream of the *Atoh1-Neurog3-NeuroD1* cascade, other transcription factors participate in specifying the differentiation into EEC sublineages. The activity of the homeodomain transcription factors *Pax4, Arx* and *Pax6* have specifically been mapped to the production of GLP+ and/or PYY+ EECs in vivo and in vitro[24,44–46]. Butyrate treatment led to a significant downregulation in *Arx* expression, with an even greater decrease observed for *Pax4* expression. In contrast, expression of the transcription factor *Pax6*, which has previously been characterised to directly modulate the expression of *NeuroD1*, was significantly upregulated by butyrate[47]. We measured a time-dependency in butyrate-mediated upregulation of both *NeuroD1* and *Pax6* transcript levels (Fig. 6c). No significant effects of acetate or propionate on the *Atoh1-Neurog3-NeuroD1* cascade were observed apart from a

**Fig. 5 | Butyrate dampens Notch activity in Hes1-GFP mouse colonic organoids.**
**a** Cartoon depicting the Notch-modulated cell fate checkpoint involved in Lgr5+ stem cell differentiation to an absorptive or secretory fate via Hes1 or Atoh1 respectively. Created in BioRender. Hirdaramani, A. (2026) https://BioRender.com/m7zws80. **b** Representative images of colonic Hes1-GFP mouse organoids in culture captured live by a widefield fluorescent microscope. Cartoons of the cellular architecture of 'budding' organoids, consisting of mature cell types and crypt-like domains populated by intestinal stem cells (ISCs) and mature cell types, and cystic organoids rich in ISCs. **c–f** Hes1-GFP organoids were monitored across a 72 h timeframe in Untreated culture conditions or cultures incubated with 2 mM of sodium butyrate (But) for 24 h. Analyses of 'budding' and 'cystic' organoids were carried out separately. Data from $n = 18$ 'budding' organoids and $n = 19$ 'cystic' organoids were collected across cultures from three separate Hes1-GFP mice. **c** Representative images of 'budding' organoids. Scale bar = 60 μm. Inset scale

bar = 25 μm. **d** Relative GFP fluorescence of individual 'budding' organoids shown as fold change of fluorescence at 0 h at each time point and GFP-negative area of individual 'budding' organoids shown as % of total organoid area. **e** Representative images of cystic organoids. Scale bar = 60 μm. Inset scale bar = 15 μm. **f** Relative GFP fluorescence and () % GFP-negative area measurements of cystic organoids respectively. Symbols represent individual organoids. (Two-way ANOVA with post-hoc Tukey's test; Untreated vs Butyrate, ****$p < 0.0001$, **$p < 0.01$, *$p < 0.05$; Untreated: 0 h vs 24 h vs 48 h vs 72 h, Butyrate: 0 h vs 24 h vs 48 h vs 72 h, ####$p < 0.0001$, ###$p < 0.001$, ##$p < 0.01$, #$p < 0.05$ for significance vs 0 h, ▲▲$p < 0.01$ for significance vs 24 h). **g** Heatmap of singscore z-scores for Hallmark pathways in Hes1-GFP organoids following 48 h or 72 h of treatment with 2 mM of sodium butyrate (But) or in untreated conditions. Organoids were derived from one Hes1-GFP mice. See also Fig. S4.

significant downregulation in *Pax6* expression by acetate (Fig. S5). Together, our findings demonstrate selective effects of butyrate on human L-cell developmental cascades that are specific to later stages of differentiation.

As we had identified a mechanistic involvement of FFAR2-Gαi activation in butyrate-mediated upregulation of PYY secretion, we next investigated the role of this pathway in modulating transcriptional programs that regulate L-cell differentiation (Fig. 6d). Neither GLPG0974 nor pertussis toxin inhibited butyrate-mediated upregulation of *NeuroD1* expression. In contrast, butyrate-mediated upregulation of *Pax6* expression was significantly depleted by both inhibitors. When we assessed the ability of AZ-1729 to modulate expression of both these targets, we observed a significant upregulation of *Pax6* but not *NeuroD1* further supporting a selective role of FFAR2-Gαi in *Pax6* expression (Fig. 6e). Dual treatment with butyrate and AZ-1729 resulted in a slight upregulation of *Pax6* expression, however this effect did not reach statistical significance ($p = 0.089$). (Fig. 6f). Together, these findings collectively support FFAR2-independent mechanisms function in regulating *NeuroD1* while the activation of an FFAR2-Gαi pathway by butyrate drives the upregulation of *Pax6* expression, and can be further enhanced pharmacologically with AZ-1729.

## Discussion

Enteroendocrine L-cells play key roles in regulation of appetite through production and secretion of anorectic gut hormones GLP-1 and PYY in response to nutrients and metabolites. In addition, the gut luminal environment is known to influence the plasticity of the gut epithelia, including the programming of enteroendocrine cell abundance and type. SCFAs are known to drive both acute gut hormone release and selectively enhance levels of PYY positive L-cells at a transcriptional and translational level[24,25] via receptor dependent and independent pathways. However, little is known about how SCFA GPCRs selectively modulate transcriptional and endocrine profiles of L-cells. In this study, we identify that butyrate promotes maturation and lineage specification of L-cells to mature PYY positive L-cells at a transcriptional, endocrine and morphological level. In human L-cell models we uncover a spatially controlled FFAR2–Gαi signalling mechanism that can be pharmacologically amplified to promote Pyy and L-cell maturation.

Although our findings confirm a marked GLP-1 dominance of the NCI-H716 hormonal repertoire by granule quantification at single-cell resolution and transcript abundance, PYY secretion was robustly detected under untreated and elevated in butyrate stimulated conditions. Previous studies that have successfully detected PYY secretion in this cell line have employed ELISAs[25,48,49], whereas radioimmunoassay-based approached have yielded undetectable signals[50]. This discrepancy might reflect differences in epitope recognition[51], peptide stability during sample processing[52] or matrix interference[53] NGS efforts in recent years have illustrated the proximal-to-distal heterogeneity of L-cells - and of all EEC subpopulations, along the human GI tract[54]; for example GCG-positive populations predominate in the ascending colon, whereas PYY-positive subpopulations are more distally enriched. Given that NCI-H716 is derived from the ascending colon, as well as immortalised, it is

therefore sensible to interpret their modelling of endogenous nutrient-derived PYY secretion with caution.

SCFAs, and in particular butyrate, are known to strongly induce a PYY-enriched hormone repertoire in human and mouse L-cells, including the NCI H716 cells employed in this study[25]. Furthermore, our data reveal a partial dependency (~50%) on FFAR2 for butyrate-induced upregulation of *Pyy* expression, consistent with the magnitude of inhibition observed in FFAR2 knockout models previously reported[25], suggesting butyrate may exert some transcriptional regulation in intestinal epithelial cells via its functionality as a HDAC inhibitor (HDACi)[55,56]. However, using FFAR2-selective antagonists and allosteric biased agonists with profiling of the upstream G protein signal pathways activated by SCFAs in human L-cells, we propose a key role for a butyrate-mediated FFAR2–Gαi signalling axis in L-cell function. Although SCFAs activate both FFAR2 and FFAR3 and both receptors were detected in NCI-H716 cells at transcript level, our data demonstrate that FFAR2 mediates Gαi- and Gαq-coupled signalling responses to acetate, propionate and butyrate in a human L-cell line. Following treatment with selective FFAR2 antagonist GLPG0974, we observed propionate to unexpectedly stimulate an increase in cAMP accumulation. Re-directing of ligand engagement between GPCRs has been reported in several systems, whereby an alternative co-expressed GPCR functions as an 'escape' receptor when the primary receptor is selectively blocked[57,58]. As such, FFAR2 antagonism may unmask or enhance propionate's ability to activate other cAMP-elevating GPCRs e.g. OR51E2 via Golf signalling[59]. This finding raises the question of whether SCFA-induced responses may become rewired in EECs under pathophysiological contexts characterised by altered FFAR2 signalling including colorectal cancer[60,61]. Consistent with findings from recombinant receptor overexpression systems[7,8], we observe a striking bias of butyrate toward Gαi activation in NCI-H716 cells, in contrast to acetate and propionate, which robustly engage both Gαi and Gαq pathways. This signalling preference may underlie the prominent FFAR2-dependent effects of butyrate on *Pyy* in human L-cells. Supporting this notion, the selective FFAR2-Gαi biased agonist AZ1729 also increased *Pyy* transcript levels, reinforcing the role of Gαi-biased signalling in this context. In addition, SCFA-activated Gαi responses exhibit varying dependence on dynamin-dependent receptor internalisation. Divergence in spatial distribution of SCFA-FFAR2-Gαi profiles (e.g. plasma membrane versus endosome), rather than absolute magnitude of response, may be an important determinant of their downstream transcriptional profiles.

Notably, our findings propose a plasma membrane-localised mechanism of a FFAR2-Gαi axis in the regulation of *Pyy* expression by butyrate and AZ1729. In contrast, our previous work identified an intracellular mode of propionate-mediated FFAR2–Gαi signalling from an endosomal compartment, which mediated acute GLP-1 secretion in mouse L-cells[11]. These findings raise the possibility that distinct SCFAs may exhibit location bias in FFAR2-Gαi signalling to diversify its actions from a common G protein signal pathway. Additionally, species-specific differences in receptor signalling may contribute to functional divergence in some L-cell responses, as evidenced by the distinct signalling and functional profiles of mouse and human FFAR2 in monocytes[62], although butyrate's ability to

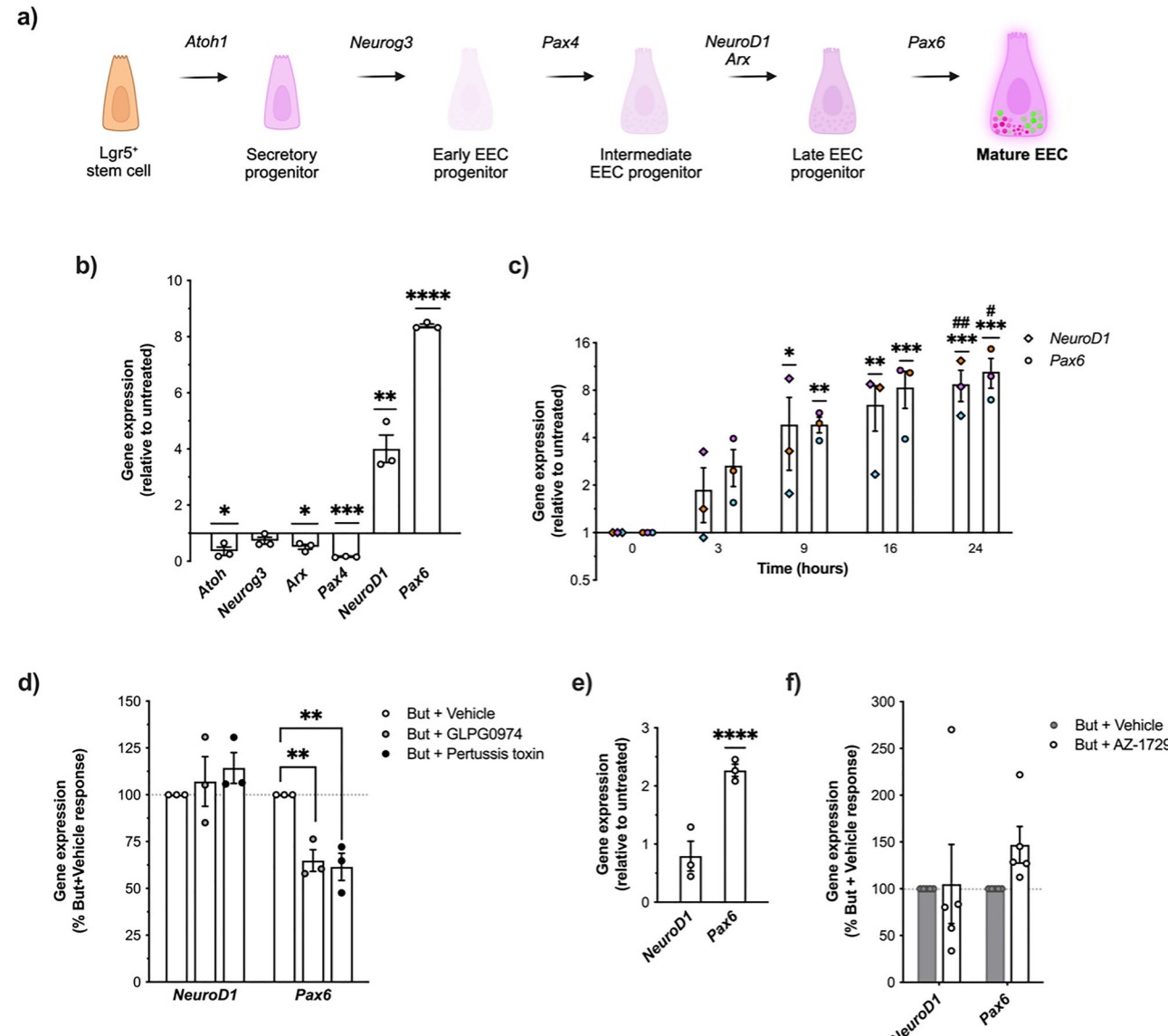

**Fig. 6 | Butyrate and AZ-1729 modulate the expression of EEC-maturation transcription factors. a** Cartoon depicting the signature transcription factor cascades involved in the differentiation of Lgr5+ stem cells to mature L-cells. Created in BioRender. Hirdaramani, A. (2026) https://BioRender.com/z3041zg. **b** Relative gene expression of transcription factors involved in L-cell differentiation (*Atoh1*, *Neurog3*, *Arx*, *Pax4*, *NeuroD1* and *Pax6*) in NCI-H716 cells treated for 24 h with 2 mM of sodium butyrate (But). Transcript levels were detected by RT-qPCR and gene expression relative to untreated was determined using the $2^{-\Delta Ct}$ method with ribosomal protein L12 (Rpl12) as a housekeeping gene control. Data are plotted as the mean ± SEM of fold-change ($2^{-\Delta\Delta Ct}$) values. Symbols depict the mean ± SEM from $n = 3$ independent experiments. Statistical significance was assessed on ΔΔCt values. (*$q < 0.05$; **$q < 0.01$; ***$q < 0.001$; ****$q < 0.0001$, *t*-test vs Untreated with post-hoc Benjamin-Hochberg multiple testing correction). **c** *NeuroD1* and *Pax6* transcript expression in NCI-H716 cells treated with 2 mM of But for 3, 9, 16 or 24 h. Transcript levels were detected by RT-qPCR and gene expression was determined as in (**b**) Data are plotted as the mean ± SEM of fold-change ($2^{-\Delta\Delta Ct}$) values. Symbols depict the mean ± SEM from $n = 3$ independent experiments, colour-coded by separate experimental runs. Statistical significance was assessed on ΔΔCt values. (One-way ANOVA with post-hoc Tukey's test; 0 h vs 3 h vs 9 h vs 16 h vs 24 h, ****$p < 0.0001$, **$p < 0.01$, *$p < 0.05$ for significance vs 0 h, ##$p < 0.01$, #$p < 0.05$ for

significance). **d** *NeuroD1* and *Pax6* transcript expression in NCI-H716 cells pre-treated with DMSO (Vehicle), FFAR2 antagonist GLPG0974 (1μM, 15 min) or Gαi inhibitor pertussis toxin (PTX) (500 ng/ml) followed by 24 h incubation with 2 mM of But. Transcript levels were detected by RT-qPCR and gene expression was determined as in (**b**). Data is shown as But + Inhibitor response as % of But + Vehicle response (dotted line at 100%) for individual gene targets. Symbols depict the mean ± SEM from $n = 3$ independent experiments (**$p < 0.001$, One-way ANOVA with post-hoc Dunnett's test vs Vehicle) **e** *NeuroD1* and *Pax6* transcript expression in NCI-H716 cells treated for 24 h with 1 μM of AZ-1729 and detected by RT-qPCR. Relative gene expression was determined as in (**b**). Data are plotted as the mean ± SEM of fold-change ($2^{-\Delta\Delta Ct}$) values data. Symbols depict the mean ± SEM from $n = 3$ independent experiments. Statistical significance was assessed on ΔΔCt values. (****$q < 0.0001$, *t*-test vs Untreated with post-hoc Benjamin-Hochberg procedure). **f** *NeuroD1* and *Pax6* transcript expression in NCI-H716 cells treated for 24 h with 2 mM But + DMSO (Vehicle) or a combination of 2 mM But and 1 μM AZ-1729 detected by RT-qPCR. Relative gene expression was determined as in (**b**). But + AZ-1729 response is shown as % of But + Vehicle response (dotted line at 100%) for individual gene targets. Symbols depict the mean ± SEM of $n = 5$ independent experiments. See also Fig. S5.

modulate differentiation in both mouse and human intestinal models in this study may indicate that certain pathways/functions may be conserved. We did observe that AZ-1729 upregulated *Pyy* expression to a smaller extent than butyrate; we propose this may be due to distinct ligand binding modes and/or conformations induced by the biased Gαi allosteric ligand compared to the orthosteric ligand butyrate from recent cryo-EM studies[63] in addition to the previously reported non-FFAR2 HDACi inhibitor properties of butyrate[25]. Butyrate also acts as a ligand to OR51E1[14]. and GPR109A[15], both of which have been detected in the NCI-H716 cell line[2,3]. However the GPR109A agonist niacin failed to upregulate Pyy transcription in this cell line[25], whereas the role of OR51E1 activation in this effect has not been investigated[64,65].

We have previously demonstrated that inulin, a fermentable substrate for SCFA production by the gut microbiome, acts via FFAR2 to selectively promote the expansion of PYY-positive, but not GLP-1-positive, L-cell populations in mice[24]. Building on these findings, our current study reveals a direct role of butyrate in influencing L-cell differentiation, selectively promoting a PYY-enriched transcriptional and functional profile. The selective effect(s) of butyrate on PYY agree with previous reports of mutually exclusive secretion of GLP-1 or PYY from L-cells upon acute nutrient challenge[28,64–67], and might reflect induction of 'switching' of hormonal repertoires of EEC lineages as they mature[44,68]. The in vivo effects of butyrate supplementation on enteroendocrine hormone secretion is conflicting. Although some studies report increases in both PYY and GLP-1 following butyrate administration[69], others have observed no significant hormonal changes[70], with outcomes also varying according to the route of delivery[71]. Since orally administered butyrate is extensively absorbed and metabolised in the proximal gastrointestinal tract - limiting its availability to the colon- the endocrine effects observed following butyrate supplementation may not faithfully recapitulate physiological responses of colonic butyrate derived via microbial fermentation. We have successfully engineered esterified propionate-fibre conjugates which ensure targeted delivery to the human colon[9]; a similar approach for butyrate may offer a more informative means of dissecting colonic butyrate-specific endocrine signalling in vivo.

Beyond its similar anorectic actions to GLP-1, PYY exerts intestinotrophic effects that support the maintenance and replenishment of the colonic epithelium[72,73]. In our findings, we observed striking morphological shift in NCI-H716 cells treated with butyrate, involving the formation of long cytoplasmic processes resembling those of PYY-positive EECs previously reported within colonic crypts[28,29]. Prior studies have demonstrated that the basal processes of PYY-expressing enteroendocrine cells (EECs), now termed neuropods, in fact resemble axonal structures both anatomically and functionally; neuropods form synapse-like connections with neurons and the intestinal milieu, enabling direct communication between the gut epithelium and the nervous system[74–78]. The existence of PYY-positive neuropods in the human colonic epithelium are yet to be characterised and might represent a unique L-cell population that is maintained by luminal and/or circulating butyrate. Utilising high intakes of fermentable fibre to increase butyrogenic bacterial populations[79] or colonic-targeted butyrate in the colon[80,81], to selectively upregulate PYY release from colonic L-cells, might add additional therapeutic benefit beyond appetite regulation in pathological contexts of epithelial injury (e.g. IBD)[82] or depleted EEC populations (e.g. obesity)[20,21]. Selective PYY release might also circumvent the incidence of adverse effects associated with GLP-1 driven appetite circuits.

SCFAs have previously been shown to transcriptionally modulate Notch-sensitive cascades of L-cell differentiation in human organoids[19]. Importantly, Notch signalling in the intestinal epithelium promotes ISC proliferation and antagonises their differentiation to secretory fates via its effector Hes1[22,33]. Using a Hes1-GFP organoid line, we tracked the effects of butyrate on Notch-mediated proliferation and inhibition of secretory lineage commitment over time at single-organoid resolution. In ISC-rich cystic organoids, butyrate induced a sustained plateau in Notch activity compared to untreated controls. In contrast, more differentiated, budding organoids displayed a progressive decline in Notch activity over time following

butyrate exposure. These dynamic patterns suggest that butyrate modulates enteroendocrine cell (EEC) differentiation in the ISC niche through two distinct mechanisms: first, by promoting the loss of stemness and initiating differentiation in immature ISCs, and second, by constraining the differentiation potential of post-mitotic progenitors and/or driving transdifferentiation of mature epithelial cells toward 'Notch-low' secretory fates. Moreover, we observed a downregulation of Notch pathway activity in butyrate-treated colonoids at the transcriptomic level, further supporting its inhibitory effect across cell types in the ISC niche.

Butyrate also modulated transcriptional programs in the human enteroendocrine cell line NCI-H716. Consistent with our organoid data, butyrate robustly upregulated expression of transcription factors (*NeuroD1* and *Pax6)* involved in the late stages of L-cell differentiation cascades that are typically suppressed by Notch signalling. Among the three major SCFAs, butyrate elicited the most pronounced transcriptional response. Moreover, we also demonstrate a concomitant transcriptional modulation of Notch-sensitive cascades involved in L-cell maturation by butyrate, and a specific dependency on FFAR2-Gαi signalling in upregulation of the late-stage transcription factor *Pax6*. In contrast, our findings indicated an FFAR2-independence in butyrate-mediated upregulation of *NeuroD1*. This mechanistic divergence might reflect stage-specific regulation; since *Pax6* precedes *NeuroD1* in L-cell transcriptional hierarchies[47], one possibility is that FFAR2-Gαi signalling preferentially influences earlier lineage-priming events in the trajectory, whereas later transcription factors are increasingly governed by butyrate's HDACi activity as sufficient nuclear accumulation has occurred. Our findings collectively support a key role for butyrate as a dietary metabolite that can influence integral transcriptional networks at distinct developmental stages, such as Notch, Wnt and YAP-TAZ, which govern ISC fate within the native stem cell niche[23] and in more mature GLP-1 L-cells to divert to a PYY-biased L-cell identity[44,68].

The FFAR2-selective allosteric modulator, AZ-1729, has been previously characterised as a biased agonist and positive allosteric modulator of FFAR2 activity for only propionate mediated actions[31,83]. Our study also reports the ability of this ligand to act as a PAM for acetate and butyrate at a second messenger level, and that its Gαi signal activity is internalisation-independent, suggesting ligand and location bias for AZ-1729. It's PAM activity enhanced butyrate-induced *Pyy* and *Pax6* expression levels, further supporting a plasma membrane FFAR2/Gαi signal pathway in promoting mature PYY positive L-cells. While employed to date as a pharmacological tool to probe the pleiotropic physiological actions of FFAR2, it also highlights the potential for such ligands to selectively amplify, or reduce, specific SCFA actions, including promoting the abundance of mature PYY⁺L-cells from the ISC niche or at a later stage of L-cell maturation to modify anorectic gut hormonal content.

Overall, we demonstrate that butyrate shapes distinct transcriptional, endocrine and morphological profiles of enteroendocrine L-cells, and that FFAR2/Gαi activity is both spatially controlled and has the potential to be selectively amplified via allosteric modulators. Despite the accelerated therapeutic application of GLP-1R agonists, approaches that could harness or modulate endogenous anorectic gut hormone responses in response to diet-induced SCFA production in a selective manner could represent efficacious anti-obesity avenues with improved gastrointestinal function.

While focusing the studies on human L-cell models afforded the availability of FFA2-selelctive antagonists, this pharmacological tool could not be applied in the Hes-1 GFP mouse model as it does not have significant affinity for mouse FFAR2[84], thus the role of butyrate in regulating NOTCH activity may be via FFAR2 dependent and/or independent mechanisms. In addition, due to lack of sensitive and selective anti-FFAR2 antibodies, a common challenge across the GPCR superfamily, analysis of receptor trafficking required expression of a FLAG-tagged receptor. Thus, while we have demonstrated a role for spatially regulated FFAR2 signalling from endogenously expressed receptor in NCI-H716 cells, we acknowledge that interpretation of endocytic profiles of human FFAR2 are from the exogenously expressed receptor.

## Materials and methods

### Experimental models

**NCI-H716 cell line culture.** NCI-H716 cells were sourced from the American Type Culture Collection (Table S1). Cells were maintained in suspension in RPMI-1640 containing 2 g/L D-glucose and 2 mM glutamine supplemented with 10% (v/v) foetal bovine serum and 100U/ml penicillin/streptomycin in a humidified incubator at 37 ÅãC with 5% $CO_2$ (Table S1). Two days before experiments, cells were seeded onto culture plates were pre-coated with Cultrex Basement Membrane Extract (Table S1). diluted to 0.5 mg/ml.

**Animals.** Organoids were established from both male and female Hes1-GFP reporter mice (Table S1) aged 19–34 weeks, with approval from IACUC/animal welfare and ethics committee of Columbia University (#AC-AABI4557). We have complied with all relevant ethical regulations for animal use. Mice were sacrificed by euthanasia with carbon dioxide asphyxiation, and death was ensured by cervical dislocation. Organoids were maintained in complete crypt culture medium, described in 'Crypt isolation & culture', or upto five passages in a humidified incubator with 5% $CO_2$ at 37 °C, with routine passaging by mechanical dissociation every 7–10 days.

**Establishment and culture of Hes1-GFP organoids.** Colons from Hes1-GFP mice were removed, flushed with ice-cold PBS and opened longitudinally[85]. After discarding the distal colon, the proximal colon was cut into 0.5 cm fragments and incubated with ice-cold PBS containing 10 mM EDTA at 4 °C for 45 min with rocking. Tissues were cleaned thoroughly with PBS and crypts were mechanically released by vigorous shaking. Isolated crypts were pelleted by centrifugation at $300 \times g$ for 5 min at 4 °C. The pellet was resuspended in growth factor-reduced Matrigel (Table S1) and crypt media, and then seeded as 25 µl domes onto flat bottom 48-well plates, which were left to solidify for 20 min at 37 °C. 250 µl of crypt media was then added on top of domes and changed every 3 days. Crypt media consisted of Advanced DMEM (supplemented with 1% Glutamax and 1% Penicillin-Streptomycin along with the following components (Table S1): 40 ng/ml Recombinant EGF, 200 ng/ml Recombinant Noggin, and 500 ng/ml Recombinant R-spondin, 1x B27 supplement, 1x N2 supplement, 3 µM CHIR99021, and 10 µM Y-27632 dihydrochloride monohydrate. Cultures were per maintained in a humidified incubator with 5% CO2 at 37 °C.

### Experimental methods

**Reagents and antibodies.** Anti-GLP-1, anti-PYY (Abcam) and anti-FLAG M1 antibodies used are listed in Table S1. Acetate, propionate and butyrate were used as sodium salts and 100 mM stocks were made up fresh in PBS. Other agonists used were AZ-1729 at 1 µM, and forskolin at 3 µM (Table S1). The inhibitors used were 3-Isobutyl-1-methylxanthine (IBMX) at 0.5 mM (5 min pre-treatment), GLPG0974 at 1 µM (15 min pre-treatment), Pertussis toxin (PTX) at 500 ng/ml (20 h pre-treatment), YM-254890 (at 10 µM (15 min pre-treatment) and Dyngo-at 50 mM (45 min pre-treatment) (Table S1).

**Plasmid construction and transfection.** FLAG-tagged human FFAR2 (FLAG-hFFAR2) was generated by PCR amplification of FFAR2 from a human GPR43 plasmid (Table S1), and ligation at EcoRV and AfeI sites of a FLAG LHR/pcDNA3.1 plasmid via digestion. FLAG-hFFAR2 was transfected into NCI-H716 cells during the seeding process as follows. FLAG-hFFAR2 was incubated with Lipofectamine 2000 (Invitrogen) for complex formation, followed by direct addition of the complex into the cellular suspension. The cells were then seeded in RPMI medium supplemented with 10% foetal bovine serum (FBS) without antibiotics and incubated for 48 h.

**RNA isolation and Real-Time Quantitative PCR (RT-qPCR).** For RNA isolation, NCI-H716 cells were seeded in 6 well plates at a density of 1.2–1.5 × 10⁶ cells/well, and RNA was isolated using TRIzol reagent (Invitrogen) as per manufacturer's protocol. Extracted RNA was treated with DNAse I (Invitrogen) to ensure removal of contaminant genomic DNA, and 1 µg was reverse transcribed to cDNA with SuperScript IV Reverse Transcriptase (Invitrogen). Target-specific primers were designed manually using National Center for Biotechnology Information Primer-BLAST Tool. Target genes were amplified (*Gcg, Pyy, Ffar2, Ffar3, Atoh1, Ngn3 NeuroD1, Pax4, Arx Pax6*), alongside *Rpl12* housekeeping gene with custom primers (Table S2), 2x PowerUp SYBR Master Mix (ThermoFisher) and nuclease-free water in a StepOne Plus Real-Time PCR system (Applied Biosystems). PCR conditions consisted of an initial biphasic incubation at 50 °C at 5 min and then 95 °C for 5 min, followed by 40 cycles of 15 s at 95 °C and 1 min at 65 °C. qPCR Samples were run in at least triplicate. PCR product specificity was confirmed by post-PCR melt-curve analysis which consisted of the following conditions: 95 °C for 15 s, 60 °C for 1 min and then incremental temperature increases by 0.3 °C every 15 s upto 95 °C. The $2^{-\Delta\Delta Ct}$ method was used calculate relative changes in gene expression[86].

For RNA isolation in mouse colonic organoids, organoids were isolated from Matrigel domes by manual dissociation and incubation with TrypLE Dissociation reagent. Cells were lysed using the QIAzol Lysis reagent (Qiagen). RNA isolation and NGS sequencing was performed at the JP Sulzberger Columbia Genome Center Core Facility.

**Fixed sample microscopy.** Confocal images of fixed samples were acquired using a Leica Stellaris 8 Inverted confocal microscope (Leica) with the LasX software. NCI-H716 were plated on 13 mm coverslips in 24- well plates at a density of $7.5 \times 10^4$/well. For receptor trafficking analyses, FLAG-hFFAR2-transfected cells were stained with FLAG M1 antibody for 30 min at 37 °C, with indicated ligands added in for the last 20 min of the incubation. If inhibitors were used, these were administered prior to M1 treatment. M1 antibody was stripped off cell membranes by washing cells three times with 0.04 M EDTA in PBS. Next, cells were fixed by incubation with 4% paraformaldehyde (PFA) in PBS for 30 min, blocked in 2% FBS for 1 h, and then permeabilised with 0.2% Triton X-100 in PBS-Ca²⁺ for 15 min. For gut hormone immunofluorescence analysis, cells were incubated with primary antibody(s) for 1 h at this point and then washed. Next, cells were incubated with secondary AlexaFluor antibodies for 1 h in the dark, washed, and mounted onto glass slides using Fluoromount-G (Invitrogen). To stain cell cytoplasm in morphology workflows, HCS CellMaskTM Deep Red (Invitrogen) was added during the last 30 min of secondary antibody incubation.

**Single granule quantification.** For quantification of GLP-1 and PYY, fixes samples wre imaged using the LIGHTNING detection module on Leica Stellaris 8, utilising a pinhole size of 0.5 Airy Units (AU) and adaptive deconvolution to maximise image resolution. Background fluorescence was corrected in FIJI (ImageJ) using the 'Subtract' function. Individual cells were manually outlined as regions of interest (ROIs). Channel-specific autothresholds were determined and consistently applied across all treatment conditions. Granules were quantified using the 'Analyse Particles' tool, with the lower limit for particle diameter set to 120 nm. Colocalisation analysis of particles was computed using the JacCoP colocalization softwareon ImageJ (Table S1).

**Morphology analysis.** For measuring morphological parameters in NCI-H716 cells, automatic particle analysis was performed using FIJI (ImageJ) in fixed cells stained with HCS CellMask™ Deep Red. To minimise observer bias, treatment allocation was blinded during the image analysis process. Binary masks were generated from single-channel images of the CellMask™ stain, with manual thresholding parameters established and maintained consistently across all experimental conditions. In more confluent samples, where cells were in close proximity or touching, the 'watershed' function was applied to separate adjacent cells. The 'Analyze Particles' function was used to identify ROIs

corresponding to individual cells, with a minimum particle area of 20 μm² set to exclude non-cellular background artifacts. The binary mask was then overlaid onto the original CellMask™ channel to verify the accurate identification of 'true' cells by the automated analysis. Quantitative measurements, including area, Feret diameter and circularity, were extracted for each cell.

**Live imaging of intracellular Ca2+ accumulation.** Intracellular $Ca^{2+}$ accumulation in live NCI-H7176 cells was captured at 1 s intervals with a SP5 inverted confocal microscope (Leica) with LasX acquisition software. NCI-H716 cells were plated on 35 mm dishes (Mattek) with 14 mm × 1.5 mm glass coverslips at a density of $8 \times 10^5$ cells/dish. Cells were incubated with Fluo-4 Direct™ calcium indicator as per manufacturer's instructions (Table S1) for 30 min in an incubator at 37 °C in the dark, and 30 min at room temperature in the dark. Fluorescent intensity was quantified in at least 50 cells per sample from using the Fiji Time series analyser plugin, and subsequently averaged across cells in each sample.

**Live imaging of Hes1-GFP organoid cultures.** Hes1-GFP organoids in culture were imaged using a widefield fluorescence microscope (Keyence). Image analysis was carried out on FIJI, whereby outlines of individual organoids, excluding the lumenal area, were manually assigned as unique ROIs. GFP fluorescence was calculated as the average pixel intensity (i.e. mean grey value) per ROI corrected to background. Total area was also computed for individual ROIs, with manual thresholding to background to determine % GFP-negative area. Image filenames were randomised and anonymised to avoid biased analysis between treatment groups.

**Whole-mount immunofluorescent staining of mouse organoids.** Organoid domes were mechanically disrupted using a blunt P1000 tips, and organoids were fixed with 4% PFA for 4 h at room temperature on a tilting platform. The fixative suspension was transferred to a low adhesion tube and organoids were allowed to settle to the bottom of the tube prior to removal of fixative. Using a blunt tip pre-coated with 0.1% BSA, organoids were carefully washed with PBS twice, and then resuspended in PBS. An 8-well Culture Slide was coated with a thin layer of Matrigel in a 1:2 dilution in media, and incubated for 5 min at 37 °C to allow the coating to partially set. Organoid suspension was carefully added dropwise to chamber slides, and slides were incubated for 90 min at 37 °C to allow the organoids to adhere. Wells were washed once with PBS, and then permeabilised in Organoid Permeabilisation Solution (0.5% Triton X-100 in PBS) for 30 min at room temperature. Next, organoids were washed once with Organoid immunofluorescence (IF) buffer (0.2% Triton X-100, 0.05% Tween in PBS), blocked in Organoid Blocking Buffer for 1 h at room temperature, and then incubated with primary antibody diluted in Organoid Blocking Buffer (% BSA in Organoid IF Buffer) at 4 °C overnight. The next day, organoids were washed 3 times with IF buffer and then incubated with Secondary antibody and Hoescht 33342 solution diluted in Organoid Blocking Buffer for 1 h at room temperature in the dark. After secondary incubation, wells were washed again. IF buffer was removed, chambers were carefully detached and ProLong Gold AntiFade Mountant (ThermoFisher, P36930) was added to the centre of each well. Coverslips were carefully lowered onto specimens and edges were sealed. Slides were imaged on a Stellaris 8 Inverted confocal microscope (Leica) using the LasX software.

**Intracellular cAMP and IP1 signalling assays.** NCI-H716 cells were plated in 96-well plates at a density of $6 \times 10^4$ cells/well and cAMP and IP1 accumulation were measured by homogeneous time-resolved fluorescence (HTRF). To measure intracellular cAMP accumulation, NCI-H716 cells were pre-treated with 0.5 mM 3-isobutyl-1- methylxanthine (IBMX) (Sigma) for 5 min prior to any ligand treatments indicated. For IP1 measurements, NCI-H716 cells were treated with indicated ligands in the presence of 50 mM LiCl. Cells were lysed in Phospho-total Lysis buffer 4 and centrifuged at $16,000 \times g$ for 15 min at room temperature. Supernatants were analysed using a HTRF cAMP Gs Dynamic Detection Kit or a HTRF IP-One Gαq Detection Kit (Table S1) as per manufacturer's instructions, and PHERAstar Plus reader (BMG LABTECH).

**PYY secretion assay.** NCI-H716 cells were grown on 24 well plates at a density of $5 \times 10^5$ cells/well and treated with Krebs buffer containing 0.2% fatty-acid free BSA and indicated ligand(s). Supernatants were collected and centrifuged at $800 \times g$ for 10 min at 4 °C, and PYY concentrations were measured by ELISA (Table S1) as per manufacturer's instructions. Cell lysates were prepared using RIPA lysis buffer and centrifuged at $16,000 \times g$ for 15 min. Lysate protein content was determined by Coomassie Blue (Bradford) assay (Thermo Fisher) and used to normalise PYY secretion values.

**RNA-seq data analysis.** Single-cell RNA-seq data from the Tabula Muris Project, generated by FACS-based full-length transcript analysis, were downloaded from the Gene Expression Omnibus (Table S1)[39]. Data were processed using the *Seurat* package (Table S1) in R[87]. Raw gene count matrices were imported and converted into Seurat objects using the CreateSeuratObject function. Cells were retained for downstream analysis if they expressed at least 500 genes and had a minimum of 50,000 unique molecular identifier counts, resulting in a dataset comprising 3950 large intestinal cells and expression profiles for 23,433 genes. Data normalisation and scaling were performed using Seurat's LogNormalize method. Cell-type annotations—epithelial cells, enterocytes, goblet cells, brush cells and enteroendocrine cells (EECs)—were assigned by integrating cell ontology metadata provided with the Tabula Muris dataset.

For bulk RNA-seq analysis, raw reads were pre-processed by the JP Sulzberger Columbia Genome Center Core Facility; transcript abundances were quantified using a *kallisto*-based pseudoalignment pipeline, and gene-level counts were obtained using the *tximport* framework. Gene-level count matrices were then imported into DESeq2. in R, filtered for inclusion of genes with >10 counts in at least 2 samples only, and normalised using DESeq2 size factor-based normalisation. Log2-transformed counts were used as input for rank-based single sample scoring by *singscore*[40]. Each sample was independently scored for enrichment against gene sets from the Hallmark collection of the MSigDB database, accessed using the *msigdbr* package[88,89]. Singscores were z-scored and then compared between treatment groups. *P*-values for singscore-derived enrichment values were obtained using null permutations, and multiple testing correction was performed using a Benjamini-Hocberg post-hoc test with false discovery rate (FDR) threshold set to 5%. Accession links and version information of all R packages used are listed in Table S1.

## Statistics and reproducibility

All statistical data analyses were performed on GraphPad Prism 9. For testing significance of a single group, a one sample *t*-test against a bounded value (e.g. 1) was performed. When comparing two groups, a two-tailed unpaired *t* test was used to determine statistical significance. For multiple *t*-tests, the Benjamini-Hochberg procedure was applied to control the FDR at 5%. For comparisons between multiple groups, a one-way ANOVA was used with Tukey's post-hoc test when comparing every mean with every other mean, or Dunnett's post-hoc test when comparing all means to a control mean. When comparing two groups under multiple conditions, a two-way ANOVA with Tukey's post-hoc test was used. For all tests, significance was considered at $p < 0.05$ or $q < 0.05$. At least three independent experimental runs were carried out per assay, with the exception of bulk RNA-seq in mouse colonoids, and each experimental repeat included at least three technical replicates per treatment condition.

## Reporting summary

Further information on research design is available in the Nature Portfolio Reporting Summary linked to this article.

## Data availability

Numerical source data from all graphs are available in Supplementary Data. Raw and processed sequencing data from this study are available from the NCBI Gene Expression Omnibus (GEO) under accession number GSE320030.

## Materials availability

FLAG-hFFAR2 construct is available from the lead contact under a materials transfer agreement with Imperial College London.

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

## Acknowledgements

This study was supported by grants from the Biological Sciences Research Council (BBSRC) and the Genesis Research Trust. We thank Professor Kevin Murphy (Imperial College London) for providing NCI-H716 cells, and Stephen Rothery at the Facility for Imaging of Light Microscopy (FILM), Imperial College London, for technical assistance with super-resolution confocal microscopy. Biorender was used for the creation of the graphical abstract and schematics.

## Author contributions

Conceptualisation, A.H., A.C.H. and G.F.; Investigation, A.H.; Resources, C.-W.C., A.C.H. and G.F.; Writing—Original Draft, A.H. and A.C.H.; Writing—Review and Editing, all authors; Visualisation, A.H.; Supervision, A.C.H. and G.F.; Funding Acquisition, all authors.

## Competing interests

The authors declare no competing interests.
