## [Transparent Peer Review file · Communications Biology]

A selective and augmentable butyrate-FFAR2 signal circuitry programs the cellular identity of enteroendocrine L-cells

Corresponding Author: Professor Aylin Hanyaloglu

Version 0:

Reviewer comments:

Reviewer #1

(Remarks to the Author)

The authors present an interesting study in L cells whereby butyrate, a FFAR2 agonist, stimulates a PYY-biased profile, an area of intense research given the interest in leveraging L cells to produce gut hormones. The manuscript is of potential interest to the readership. However, I recommend that the following points be addressed to enhance clarity and interpretation.

While the manuscript addresses an important topic, one area where clarity could be improved is in the visualization of the data. Currently, the figures rely heavily on summary statistics and error bars, which can obscure the distinction between biological replicates and within-sample variability. Given that the authors performed multiple experiments and replicates, I encourage the authors to consider incorporating approaches such as those described by Lord et al. (J Cell Biol, 2020, "SuperPlots: Communicating reproducibility and variability in cell biology"), which provide a clear framework for simultaneously displaying cell-level data and experimental reproducibility. This would enhance transparency and enable readers to assess and interpret the findings with greater confidence.

Some previous findings in NCI-H716 cells (J Mol Endocrinol. 2016 Jan 27;56(3):201–211. doi: 10.1530/JME-15-0293) reported that Peptide YY, which normally co-localizes with GLP-1 in distal L-cells, was not detected in any of the tested cell lines. Given this discrepancy, it would strengthen the manuscript if the authors could discuss how their current methods/findings align or contrast with those prior results, and what this may imply for interpretation.

The data in 4C only appear to be 2 replicates, and the axis starts at 1, and the statistics suggest the differences are defined as " $p < 0.05$, One-sample Control value (1) 396 vs Ligand(s)" which is less clear for the readers. Additionally, the spread of the three replicates in Figure 6F is quite large, making interpretation of the results difficult. While the pattern may be consistent with Pax6, the variability diminishes confidence in the conclusion. Clarification of this analysis or additional supporting data would be helpful, given the conclusions drawn regarding az-1729.

The analysis of Hes1 expression using single-cell data from Tabula Muris is valuable. However, the rationale for using HES1-GFP as a surrogate for differentiation in both types of organoids should be strengthened. The current interpretation relies heavily on this marker, and additional justification would enhance the robustness of the conclusions.

Reviewer #2

(Remarks to the Author)

This study by Hirdaramani and colleagues explores the relationship between butyrate and PYY hormone production and secretion through FFAR2 and Notch signaling. Through the use of both NCI-H716 cells and mouse colonoids, they show butyrate induces FFAR2 activation through Gai signaling to increase PYY expression and secretion independent of internalization. Hirdaramani and colleagues then show that butyrate decreases Hes1 expression using a Hes1-GFP mouse colonoid model and increases expression of NeuroD1 and Pax6, but possibly through different mechanisms. This study build on previous published studies from this group, which explore how different short chain fatty acids regulate enteroendocrine hormone production and secretion. It describes a very unique regulatory mechanism driving PYY production, which is very useful to the GI field with the growing concerns around obesity and its negative health sequelae. However, many of the conclusions drawn from this paper are overstated and would benefit from multiple confirmatory

studies.

Major points:

1. A conclusion is made regarding the shape of L-cells in the colon compared to butyrate-treated NCI-H716 cells. To make this claim, there should be a comparison of L-cell morphology from human tissues.
2. If possible, using a FFAR3 antagonist to further show that these SCFAs signal through FFAR2 to drive Gai responses would be helpful.
3. In reviewing Figure 2, it appears that butyrate does not signal much through Gaq and signals less through Gai compared to pro and ace. Are there any other mechanisms that butyrate could be signaling through and can they be evaluated?
4. In figure 4C, AZ-1729 does not increase Pyy expression as much as butyrate alone, suggesting an additional mechanism outside of FFAR2 signaling. However, this is not discussed. Is it possibly binding to other receptors, like GPR109A? If yes, can this be proven?
5. Figure 5 suggests butyrate inhibits Notch activity; however, I do not believe this statement can be made solely through Hes1 expression. Additional studies should be performed to show Notch inhibition.
6. Figure 6 is compelling, but would benefit from similar studies being performed in the mouse colonoids.
7. It is unclear why a one-tailed t-test was used for their analyses. Further, multiple experiments appear to use multiple t-tests, in which case they should be corrected for multiple comparisons.

Minor points:

1. In Figure 1E, there is no significant increase in %PYY granule pool compared to untreated, suggesting that butyrate does not increase PYY granule numbers in these cells.
2. Letter designations should be used for the supplemental figures. For example, Figure S1A instead of S1.
3. The first paragraph in the results section is confusing, with multiple sentences in contradiction to the figures. For example, "SCFA-mediated increases in gene expression of Gcg and Pyy were confirmed (Figure 1A)." However, figure 1A shows decreased expression in Pro and But.
4. Figure 1C legend states there is an inset scale bar which does not exist in the figure.
5. There is no discussion regarding propionate inducing an increased cAMP signaling following treatment of GLPG0974, which appears to be a very unique finding
6. In figure 4C, there appears to only be an N of 2 for AZ-1729 and But + AZ; this experiment should be repeated with an N of 3.
7. In figure 5, can you please show the significance across time in a different graph?
8. There needs to be more discussion regarding the difference in Neurod1 and Pax6 regulation. Why is Neurod1 expression not altered by GLPG0974, PTX, and AZ1729, but Pax6 is?

Reviewer #3

(Remarks to the Author)

In this study by Hirdaramani et al, authors uncover the molecular mechanisms by which known regulators of gut hormone secretion butyrate and a selective FFAR2 agonist influence the L-cell differentiation and how this is reflected in their hormone-secreting capacity. This is an interesting study which also makes an attempt to develop a pharmacologically sound approach to modulate L cell differentiation in the gut. The paper is very well-written, the findings are novel and of interest to scientists in the field of pharmacology in metabolic diseases, gut targeting and enteroendocrine cells. The experiments are well-executed and conclusions are well substantiated.

Questions to the authors:

1. Authors find that butyrate, known and promising therapeutic agent, favours PYY production by the L-cells. Could authors add to the discussion which physiological implications would result from this particular property? Would it be possible to also mention whether enhanced PYY secretion rather than GLP-1 secretion can be observed in vivo with butyrate supplementation?

2) The data from morphological analyses induced by butyrate and other SCFAs in NCI-H716 cells pointed to an important action of butyrate which could not have been observed in another cell platform and thus lead to further investigation. Did you also observe changes in the crypt length in organoids?

Version 1:

Reviewer comments:

Reviewer #1

(Remarks to the Author)

The authors have addressed my concerns.

Reviewer #2

(Remarks to the Author)

The authors have been thoroughly responsive, and the manuscript is considerably improved with new data and discussion.

1. I still have some concerns regarding Figure 6, which I believe would benefit from ex vivo organoid analysis and be a natural progression from Figure 5. There are many studies that show differentiation of EE cells in murine organoids is possible, so it is unclear to me why murine organoids cannot be treated with butyrate or AZ-1729 and then evaluated for EE cell differentiation using qPCR. If GLPG0974 does not work on mouse cells, does pertussis toxin?

Reviewer #3

(Remarks to the Author)

Authors have fully addressed all the issues raised in my review. I have no further questions. My recommendation is to publish.

Dear Reviewer(s),

Thank you for taking the time to review our paper, and for the valuable feedback. Please see below for how we have addressed each revision point:

Reviewer #1

'While the manuscript addresses an important topic, one area where clarity could be improved is in the visualization of the data. Currently, the figures rely heavily on summary statistics and error bars, which can obscure the distinction between biological replicates and within-sample variability. Given that the authors performed multiple experiments and replicates, I encourage the authors to consider incorporating approaches such as those described by Lord et al. (J Cell Biol, 2020, "SuperPlots: Communicating reproducibility and variability in cell biology"), which provide a clear framework for simultaneously displaying cell-level data and experimental reproducibility. This would enhance transparency and enable readers to assess and interpret the findings with greater confidence.'

Superplots have been added to figure 1 & 2, depicting the mean of each experimental repeat as larger differently coloured symbols, and individual cell values as smaller symbols. Where biological variability is particularly high, means of individual repeats have been coloured to highlight key trends between treatment conditions (figure 3, 4 & 6).

'Some previous findings in NCI-H716 cells (J Mol Endocrinol. 2016 Jan 27;56(3):201–211. doi: 10.1530/JME-15-0293) reported that Peptide YY, which normally co-localizes with GLP-1 in distal L-cells, was not detected in any of the tested cell lines. Given this discrepancy, it would strengthen the manuscript if the authors could discuss how their current methods/findings align or contrast with those prior results, and what this may imply for interpretation.'

The following text has been added to the Discussion:

Although our findings confirm a marked GLP-1 dominance of the NCI-H716 hormonal repertoire by granule quantification at single-cell resolution and transcript abundance, PYY secretion was robustly detected under untreated and elevated in butyrate stimulated conditions. Previous studies that have successfully detected PYY secretion in this cell line have employed ELISAs [1–3], whereas radioimmunoassay-based approaches have yielded undetectable signals [4]. This discrepancy might reflect differences in epitope recognition [5], peptide stability during sample processing [6] or matrix interference [52]. NGS efforts in recent years have illustrated the proximal-to-distal heterogeneity of L-cells - and of all EEC subpopulations, along the human GI tract [8]; for example GCG-positive populations predominate in the ascending colon, whereas PYY-positive subpopulations are more distally enriched. Given that NCI-H716 is derived from the ascending colon, as well as immortalised, it is therefore sensible to interpret their modelling of endogenous nutrient-derived PYY secretion with caution.

*'The data in 4C only appear to be 2 replicates, and the axis starts at 1, and the statistics suggest the differences are defined as $**p < 0.05$, One-sample Control value (1) 396 vs Ligand(s)" which is less clear for the readers.'*

The y-axis upper limit for fig 4C. has been amended so that the mean of all 3 replicates is visible. Statistical analysis for fig 4C. has been changed (and figure legend updated) to a one-way ANOVA with Dunnett's multiple comparisons.

'Additionally, the spread of the three replicates in Figure 6F is quite large, making interpretation of the results difficult. While the pattern may be consistent with Pax6, the variability diminishes confidence in the conclusion. Clarification of this analysis or additional supporting data would be helpful, given the conclusions drawn regarding az-1729.'

A further two biological repeats were carried out for figure 6F.

'The analysis of Hes1 expression using single-cell data from Tabula Muris is valuable. However, the rationale for using HES1-GFP as a surrogate for differentiation in both types of organoids should be strengthened. The current interpretation relies heavily on this marker, and additional justification would enhance the robustness of the conclusions.'

To address the concerns raised, we performed bulk RNAseq analysis of mouse colonoids treated with/without butyrate for either 48 or 72h. Using the singscore method to compute enrichment of HallMark pathways from the Msigdb, we confirmed a significant enrichment of Notch Signalling in butyrate-treated cultures at 48h and 72h.

Reviewer #2

'A conclusion is made regarding the shape of L-cells in the colon compared to butyrate-treated NCI-H716 cells. To make this claim, there should be a comparison of L-cell morphology from human tissues.'

Prior studies have reported the elongated morphology via EM-based approaches (cited in Results section in this manuscript), and we report a similar observation in vitro following butyrate treatment. The human L-cell model has enabled us to observe morphological changes with butyrate due to the immature nature of this L-cell line. An intestinal crypt sample would be challenging to track L-cells over time following butyrate treatment as they are in low abundance and maintaining integrity in culture is limited temporally. We have instead modified the text so we do not overstate our conclusions, ensuring these are confined to the cell lines employed, however, we do draw parallels to observations made by others for the capacity of L-cells to exhibit this morphological profile.

The following text has been added to the Results:

In addition, NCI-H716 appeared to undergo a prominent morphological shift upon treatment with butyrate, exhibiting elongated morphologies characteristic of specific L-cell populations enriched in the human distal intestine [9,10] (Figure 1f). PYY-immunoreactive cells in primary human colonic tissue have been shown to often adopt a tall anatomy with extended cytoplasmic processes, weaving between cells from the lamina to the gut lumen [9,10].

'If possible, using a FFAR3 antagonist to further show that these SCFAs signal through FFAR2 to drive Gai responses would be helpful.'

Unfortunately, FFAR3 antagonists are not commercially available. The academic group that has previously reported on the FFAR3 antagonist AR399519 have disclosed concerns that this compound exhibits agonist-like behaviour in GTP γ S assays (personal communication). We have confirmed that despite the detectable, but very low, mRNA expression of FFAR3, there are no functional FFAR3 responses in NCI-H716 cell line by using the selective FFAR3 agonist TUG-1907 [11] (kind gift from Drs. Elisabeth Ulven and Trond Ulven). We demonstrate that TUG-1907 does not activate Gi signalling at doses up to 100 μ M. Combined with our findings that the FFAR2-selective PAM, AZ-1729, promotes butyrate-mediated signalling and downstream functions, and an FFAR2-selective antagonist inhibits Gi signaling from all three SCFAs, support a role for only functional FFAR2 in this cell model. Please see figure below.

TUG-1907

In reviewing Figure 2, it appears that butyrate does not signal much through Gαq and signals less through Gαi compared to pro and ace. Are there any other mechanisms that butyrate could be signaling through and

can they be evaluated?'

The reviewer is correct and this is in line with prior reports from others for human FFAR2 overexpressed in heterologous cell lines [12,13]. We have demonstrated that the butyrate-driven transcriptional changes require Gi, and not Gq signalling, in addition that this signalling does not require receptor internalization. Although the Gi signal from butyrate is not as robust as propionate or acetate, our data suggest, along with increasing reports of location bias in GPCR signalling, that location over signal strength, may direct downstream cellular functions.

The following text has been added to the Discussion for clarity:

In addition, SCFA-activated Gαi responses exhibit varying dependence on dynamin-dependent receptor internalisation; divergence in spatial distribution of SCFA-FFAR2-Gai profiles (e.g plasma membrane via endosome), rather than absolute magnitude of response, may be an important determinant of their downstream transcriptional profiles.

'In figure 4C, AZ-1729 does not increase Pyy expression as much as butyrate alone, suggesting an additional mechanism outside of FFAR2 signalling. However, this is not discussed. Is it possibly binding to other receptors, like GPR109A? If yes, can this be proven?'

We thank the reviewer for highlighting this interesting feature of the allosteric modulator.

The following text has been added to the Discussion:

We did observe that AZ-1729 upregulated Pyy expression to a smaller extent than butyrate; we propose this may be due to distinct ligand binding modes and/or conformations induced by the biased Gai allosteric ligand compared to the orthosteric ligand butyrate from recent cryo-EM studies [63] in addition to the previously reported non-FFAR2 HDACi inhibitor properties of butyrate [25].

Butyrate also acts as a ligand to OR51E1 [14] and GPR109A [15], both of which have been detected in the NCI-H716 cell line [2,3]. However the GPR109A agonist niacin failed to upregulate Pyy transcription in this cell line [3], whereas the role of OR51E1 activation in this effect has not been investigated. [64][65]

'Figure 5 suggests butyrate inhibits Notch activity; however, I do not believe this statement can be made solely through Hes1 expression. Additional studies should be performed to show Notch inhibition.'

Please refer to Reviewer #1 point 4

'Figure 6 is compelling, but would benefit from similar studies being performed in the mouse colonoids.' –

We have acknowledged in the limitations section that the FFAR2-selective antagonist GLPG0974 does not effectively block the mouse receptor ortholog [14,15] and thus could not be used in the mouse colonoid studies of Figure 5. The Hes1-GFP mouse line was used for its ability to report Notch activity at single organoid resolution, and allowed us to separately investigate the effects of butyrate at earlier (in cystic populations) and later (in budding populations) developmental stages of EEC differentiation.

'It is unclear why a one-tailed t-test was used for their analyses. Further, multiple experiments appear to use multiple t-tests, in which case they should be corrected for multiple comparisons.'

Statistical analyses have been amended. For comparison between two conditions, statistical analyses have been amended to two-tailed t-tests. For multiple t-tests, the Benjamin-Hochberg procedure with FDR correction <5% was applied.

'In Figure 1E, there is no significant increase in %PYY granule pool compared to untreated, suggesting that butyrate does not increase PYY granule numbers in these cells.'

Please refer back to Figure 1E – this effect is significant. We have also amended Figure 1d to a multivariable plot for clarity; this shows GLP-1 and PYY granule number per cell across repeats. The upregulation of PYY granules by butyrate is also significant.

'Letter designations should be used for the supplemental figures. For example, Figure S1A instead of S1.'

This has been amended.

'The first paragraph in the results section is confusing, with multiple sentences in contradiction to the figures. For example, "SCFA-mediated increases in gene expression of Gcg and Pyy were confirmed (Figure 1A)." However, figure 1A shows decreased expression in Pro and But.'

This has been amended.

'Figure 1C legend states there is an inset scale bar which does not exist in the figure.'

This has been amended.

'There is no discussion regarding propionate inducing an increased cAMP signaling following treatment of GLPG0974, which appears to be a very unique finding.'

The following text has been added to the Discussion:

Following treatment with selective FFAR2 antagonist GLPG0974, we observed propionate to unexpectedly stimulate an increase in cAMP accumulation. Redirecting of ligand engagement between GPCRs has been reported in several systems, whereby an alternative co-expressed GPCR functions as an 'escape' receptor when the primary receptor is selectively blocked [16,17]. As such, FFAR2 antagonism may unmask or enhance propionate's ability to activate other cAMP-elevating GPCRs e.g., OR51E2 via Golf signalling [18]. This finding raises the question of whether SCFA-induced responses may become rewired in EECs under pathophysiological contexts characterised by altered FFAR2 signalling including colorectal cancer [19,20].

'In figure 4C, there appears to only be an N of 2 for AZ-1729 and But + AZ; this experiment should be repeated with an N of 3.'

Please refer to Reviewer #1 point 2.

'In figure 5, can you please show the significance across time in a different graph?'

This has been amended.

'There needs to be more discussion regarding the difference in Neurod1 and Pax6 regulation. Why is Neurod1 expression not altered by GLPG0974, PTX, and AZ1729, but Pax6 is?'

The following text has been added to the Discussion:

In contrast, our findings indicated an FFAR2-independence in butyrate-mediated upregulation of *NeuroD1*. This mechanistic divergence might reflect stage-specific regulation; since *Pax6* precedes *NeuroD1* in L-cell transcriptional hierarchies [21], one possibility is that FFAR2-Gi signalling preferentially influences earlier lineage-priming events in the trajectory, whereas later transcription factors are increasingly governed by butyrate's HDACi activity as sufficient nuclear accumulation has occurred.

Reviewer #3:

‘Authors find that butyrate, known and promising therapeutic agent, favours PYY production by the L-cells. Could authors add to the discussion which physiological implications would result from this particular property? Would it be possible to also mention whether enhanced PYY secretion rather than GLP-1 secretion can be observed in vivo with butyrate supplementation?’

To address your first point, the following text has been added to the discussion:

Utilising butyrate, and/or high intakes of fermentable fibre to increase butyrogenic bacterial populations [22] or butyrate flux in the colon [23,24], to selectively upregulate PYY release from colonic L-cells, may add additional therapeutic benefit beyond appetite regulation in pathological contexts of epithelial injury (e.g IBD) [25] or depleted EEC populations (e.g obesity) [26,27]. Selective PYY release might also circumvent the incidence of adverse effects associated with GLP-1 driven appetite circuits.

To address your second point, the following text has been added to the discussion:

The *in vivo* effects of butyrate supplementation on enteroendocrine hormone secretion is conflicting. Although some studies report increases in both PYY and GLP-1 following butyrate administration [69], others have observed no significant hormonal changes [70], with outcomes also varying according to the route of delivery [71]. Since orally administered butyrate is extensively absorbed and metabolised in the proximal gastrointestinal tract - limiting its availability to the colon - the endocrine effects observed following butyrate supplementation may not faithfully recapitulate physiological responses of colonic butyrate derived via microbial fermentation. We have successfully engineered esterified propionate-fibre conjugates which ensure targeted delivery to the human colon [9]; a similar approach for butyrate may offer a more informative means of dissecting colonic butyrate-specific endocrine signalling *in vivo*.

‘The data from morphological analyses induced by butyrate and other SCFAs in NCI-H716 cells pointed to an important action of butyrate which could not have been observed in another cell platform and thus lead to further investigation. Did you also observe changes in the crypt length in organoids?’

Total area measurements of cystic and budding organoids are indicated in figure S4b and S4c respectively – we did not observe significant changes in total area with butyrate treatment, however % GFP-negative areas were significantly affected (figure 5d, 5f).

1. Kim, K. S., Egan, J. M. & Jang, H. J. Denatonium induces secretion of glucagon-like peptide-1 through activation of bitter taste receptor pathways. *Diabetologia* 57, 2117–2125 (2014).
2. Han, Y. E. *et al.* Olfactory Receptor OR51E1 Mediates GLP-1 Secretion in Human and Rodent Enteroendocrine L Cells. *J Endocr Soc* 2, 1251–1258 (2018).
3. Larraufie, P. *et al.* SCFAs strongly stimulate PYY production in human enteroendocrine cells. *Sci Rep* 8, (2018).
4. Kuhre, R. E. *et al.* Peptide production and secretion in GLUTag, NCI-H716, and STC-1 cells: A comparison to native L-cells. *J Mol Endocrinol* 56, 201–211 (2016).
5. Albrechtsen, N. J. W. & Rehfeld, J. F. On premises and principles for measurement of gastrointestinal peptide hormones. *Peptides (N.Y.)* 141, 170545 (2021).
6. Yi, J., Warunek, D. & Craft, D. Degradation and Stabilization of Peptide Hormones in Human Blood Specimens. *PLoS One* 10, e0134427 (2015).
7. Tate, J. & Ward, G. Interferences in immunoassay. *Clin Biochem Rev* 25, 105–20 (2004).
8. Burclaff, J. *et al.* A Proximal-to-Distal Survey of Healthy Adult Human Small Intestine and Colon Epithelium by Single-Cell Transcriptomics. *CMGH* 13, 1554–1589 (2022).
9. Böttcher, G. *et al.* Coexistence of peptide YY and glicentin immunoreactivity in endocrine cells of the gut. *Regul Pept* 8, 261–266 (1984).
10. El-Salhy, M. *et al.* Immunocytochemical identification of polypeptide YY (PYY) cells in the human gastrointestinal tract. *Histochemistry* 77, 15–23 (1983).
11. Ulven, E. R. *et al.* Structure–Activity Relationship Studies of Tetrahydroquinolone Free Fatty Acid Receptor 3 Modulators. *J Med Chem* 63, 3577–3595 (2020).
12. Le Poul, E. *et al.* Functional characterization of human receptors for short chain fatty acids and their role in polymorphonuclear cell activation. *Journal of Biological Chemistry* 278, 25481–25489 (2003).
13. Nilsson, N. E., Kotarsky, K., Owman, C. & Olde, B. Identification of a free fatty acid receptor, FFA2R, expressed on leukocytes and activated by short-chain fatty acids. *Biochem Biophys Res Commun* 303, 1047–1052 (2003).
14. Sergeev, E. *et al.* A single extracellular amino acid in Free Fatty Acid Receptor 2 defines antagonist species selectivity and G protein selection bias. *Sci Rep* 7, 13741 (2017).
15. Sergeev, E. *et al.* Non-equivalence of Key Positively Charged Residues of the Free Fatty Acid 2 Receptor in the Recognition and Function of Agonist Versus Antagonist Ligands. *Journal of Biological Chemistry* 291, 303–317 (2016).
16. Hannan, R. E., Davis, E. A. & Widdop, R. E. Functional role of angiotensin II AT 2 receptor in modulation of AT 1 receptor-mediated contraction in rat uterine artery: involvement of bradykinin and nitric oxide. *Br J Pharmacol* 140, 987–995 (2003).
17. Tachezy, M. *et al.* CXCR7 expression in esophageal cancer. *J Transl Med* 11, 238 (2013).
18. Pluznick, J. A novel SCFA receptor, the microbiota, and blood pressure regulation. *Gut Microbes* 5, 202–207 (2014).

19. Sivaprakasam, S. *et al.* An essential role of Ffar2 (Gpr43) in dietary fibre-mediated promotion of healthy composition of gut microbiota and suppression of intestinal carcinogenesis. *Oncogenesis* 5, e238–e238 (2016).
20. Al Mahri, S. *et al.* Free fatty acids receptors 2 and 3 control cell proliferation by regulating cellular glucose uptake. *World J Gastrointest Oncol* 12, 514–525 (2020).
21. Marsich, E., Vetere, A., Di Piazza, M., Tell, G. & Paoletti, S. *The PAX6 Gene Is Activated by the Basic Helix-Loop-Helix Transcription Factor NeuroD/BETA2.* *Biochem. J* vol. 376 (2003).
22. Van-Wehle, T. & Vital, M. Investigating the response of the butyrate production potential to major fibers in dietary intervention studies. *NPJ Biofilms Microbiomes* 10, 63 (2024).
23. Baxter, N. T. *et al.* Dynamics of Human Gut Microbiota and Short-Chain Fatty Acids in Response to Dietary Interventions with Three Fermentable Fibers. *mBio* 10, (2019).
24. Mayorga-Ramos, A., Barba-Ostria, C., Simancas-Racines, D. & Guamán, L. P. Protective role of butyrate in obesity and diabetes: New insights. *Front Nutr* 9, (2022).
25. Roda, G. Intestinal epithelial cells in inflammatory bowel diseases. *World J Gastroenterol* 16, 4264 (2010).
26. Wölnerhanssen, B. K. *et al.* Deregulation of transcription factors controlling intestinal epithelial cell differentiation; a predisposing factor for reduced enteroendocrine cell number in morbidly obese individuals. *Sci Rep* 7, 8174 (2017).
27. Osinski, C. *et al.* Type 2 diabetes is associated with impaired jejunal enteroendocrine GLP-1 cell lineage in human obesity. *Int J Obes* 45, 170–183 (2021).

Dear Reviewer(s),

Thank you for taking the time to review our paper, and for the valuable feedback. Please see below for how we have addressed your point of concern:

'I still have some concerns regarding Figure 6, which I believe would benefit from ex vivo organoid analysis and be a natural progression from Figure 5. There are many studies that show differentiation of EE cells in murine organoids is possible, so it is unclear to me why murine organoids cannot be treated with butyrate or AZ-1729 and then evaluated for EE cell differentiation using qPCR. If GLPG0974 does not work on mouse cells, does pertussis toxin?'

We do not see pertussis toxin as a suitable tool for mechanistically modelling butyrate-FFAR2-Gai due to its non-selective nature amongst Gi/o-coupled GPCRs. Moreover, EE cells comprise <1% of the intestinal stem cell niche, and their low population abundance in organoid cultures without transgenic modification is representative of this. As such, EE cell-restricted effects bulk organoids analyses can be masked by low signal:noise ratios. We have performed an enrichment analysis using a mouse EEC differentiation gene set with our existing bulk RNA-seq data (see scores below) which did not capture a marked effect of butyrate on EEC cell expansion; this is likely due to the small fraction of EE cells present in bulk samples, as well as the highly transient and plastic nature of cellular maturation (global scores decline at 72h). Importantly, we do demonstrate an upregulation of PYY+ cells in butyrate-treated organoid cultures by immunofluorescence staining, which we have added to Supplementary Figure 4.

	EEC differentiation z-Score
Untreated_48	0.8444839
Butyrate_48	0.8799032
Untreated_72	-0.975908
Butyrate_72	-0.7484791